# LogicGuard: Improving embodied LLM agents through temporal logic based critics

## Abstract

Large language models (LLMs) have shown promise in zero-shot and single step reasoning and decision-making problems, but in long-horizon sequential planning tasks, their errors compound, often leading to unreliable or inefficient behavior. We introduce LogicGuard, a modular actor–critic architecture in which an LLM actor is guided by a trajectory-level LLM critic that communicates through Linear Temporal Logic (LTL). Our setup combines the reasoning strengths of language models with the guarantees of formal logic. The actor selects high-level actions from natural language observations, while the critic analyzes full trajectories offline and generates LTL constraints. An online verifier enforces these constraints, blocking unsafe or inefficient actions before execution. LogicGuard supports both fixed safety rules and adaptive, learned constraints, and is model-agnostic: any LLM-based planner can serve as the actor, with LogicGuard acting as a logic-generating wrapper. We formalize planning as graph traversal under symbolic constraints, allowing LogicGuard to analyze failed or suboptimal trajectories and generate new temporal logic rules that improve future behavior. To demonstrate generality, we evaluate LogicGuard across two distinct settings: short-horizon general tasks and long-horizon specialist tasks. On the Behavior benchmark of 100 household tasks, LogicGuard increases task completion rates by 25% over a baseline InnerMonologue planner. On the Minecraft diamond-mining task, which is long-horizon and requires multiple interdependent subgoals, SayCan fails to complete the task without LogicGuard, while LogicGuard-enabled SayCan succeeds consistently. Moreover, LogicGuard applied to InnerMonologue yields a 23% efficiency gain over vanilla InnerMonologue. These results show that enabling LLMs to supervise each other through temporal logic yields more reliable, efficient and safe decision-making.

## 1 Introduction

Large Language Models (LLMs) have recently demonstrated strong performance on diverse reasoning and decision-making tasks, including natural language understanding, reasoning (Huang & Chang, 2022; Wei et al., 2022), and code generation (Li et al., 2022b; Chen et al., 2021). However, much of this success has been in static, text-based settings. In contrast, embodied dynamical environments require agents to plan over long horizons under uncertainty, partial observability, and complex dynamics. While LLMs can generate short-term plans or respond coherently to individual prompts, they lack the consistency, memory, and iterative refinement needed to solve multi-step tasks where intermediate actions must align with long-term goals. These shortcomings are especially pronounced in open-ended domains such as robotics and interactive virtual environments, where agents must reason over large action spaces, tools, and environment dynamics.

Recent evaluations (Kambhampati et al., 2024) show that even in simplified settings, LLMs often fail to produce reliable plans without external verification. Small prompt variations lead to compounding errors, and generated plans frequently violate preconditions or logical constraints. To address these weaknesses, hybrid frameworks pair LLMs with external verifiers (Silver et al., 2022; Guo et al., 2024). Yet, these methods typically rely on manual design and labeling, limiting scalability. In this work, we aim to reduce such manual effort by automatically generating formal constraints that guide and safeguard LLM planning.

Embodied environments such as Minecraft (Fan et al., 2022; Wang et al., 2023a), iGibson (Li et al., 2022a), and VirtualHome (Puig et al., 2018) serve as useful testbeds for developing such architectures, while datasets like Behavior (Li et al., 2023) benchmark agent performance across diverse tasks. These domains capture core challenges of embodied intelligence: (i) perceiving high-dimensional states,(ii) planning under sparse supervision, and (iii) executing long-horizon strategies. Recent work applies LLMs in these settings, from household robotics (Ahn et al., 2022) to resource-driven virtual worlds (Wang et al., 2023a), but scalability and reliability remain open problems.

Ultimately, enabling LLMs to function as trustworthy autonomous agents requires robustness in safety-critical, long-horizon, and multi-agent contexts such as healthcare (Hosny et al., 2018), automated transportation (Wang et al., 2023b), and domestic assistance (Birkmose et al., 2025). In these domains, unsafe or inconsistent behavior risks physical failures, while inefficiency erodes human trust (Esterwood & Robert Jr, 2023). We argue that planning architectures must combine the flexible reasoning of LLMs with the formal guarantees of symbolic logic. To this end, we propose a symbolic actor-critic framework that uses temporal logic to enforce safety, improve performance, and ensure interpretable decision-making in embodied environments.

## 1.1 RELATED WORKS

**Temporal Logic for Planning and Reinforcement learning** Linear Temporal Logic (LTL) (Pnueli, 1977) is a formal language for specifying temporal properties of systems through boolean logic based expressions. LTL has been widely used in robot motion planning (Fainekos et al., 2005), for safe planning and control (Wongpiromsarn et al., 2012), and even in reinforcement learning (Alshiekh et al., 2018). In each of these applications, LTL is used to specify safety constraints; instead, we shall use it to specify performance constraints.

**Language Models for Planning and Policy Learning** Recent work has explored LLMs for short-horizon planning (Huang et al., 2022a). SayCan (Ahn et al., 2022) scores actions with an LLM and weights them by an affordance function, while InnerMonologue (Huang et al., 2022b) feeds LLM feedback back into itself to enable online re-planning. In embodied environments, Ziliotto et al. (2025) use compositional planning for LLMs, while Wu et al. (2023) use alignment to improve reasoning in embodied environments. Chen et al. (2024) breaks down complicated tasks into subgoals. Other works such as (Kambhampati et al., 2024) suggest that LLMs do not show reliability while operating autonomously, and require external critics. Our work builds directly upon these ideas, augmenting LLMs with formal verifiers.

**Integrating Symbolic Reasoning with LLMs** Recent work at the intersection of LLMs and temporal logic focuses on translating natural language into LTL constraints (Chen et al., 2023; Liu et al., 2023), or enforcing such translations during execution to filter unsafe actions (Yang et al., 2024).These methods use static handcrafted rules. Recent work (Ravichandran et al., 2025) uses LLMs to generate safety constraints online for robotics, however their focus is purely on avoiding unsafe behaviors. Further, Manas et al. (2024) proposes a conversion mechanism from specific natural language planning instructions or goal descriptions to formalized LTL constraints. More recent work Lee et al. (2025) is also able to infer safety rules from vague task descriptions. In our work, we use LTL not only as a means of satisfying predefined safety and goal constraints, but also as a dynamic tool to improve planning efficiency. We leverage an LLM-based LTL law generator to actively improve planning and long-horizon task performance, enabling adaptive, empirically grounded constraint generation in complex sequential environments.

**LLM-guided Actor-Critic Architectures** Prior literature explores hybrid actor-critic setups where LLMs guide planning and evaluation, using natural language feedback or prompt optimization loops (Dong et al., 2023; Yang et al., 2025). Recent advances in this line of work have lead to a subclass of architectures termed "LLM-as-a-judge" (Li et al., 2025; Khan et al., 2025), which employ LLMs to evaluate other algorithms via numeric scores or natural-language feedback. Due to their reliance on opaque numeric scores or vague natural language feedback, these architectures provide no formal guarantees, which are especially crucial in multi-step embodied reasoning tasks. Our use of LTL as a formal language of communication offers interpretable, precise, symbolic evaluation enforcing both safety and performance constraints.

## 1.2 CONTRIBUTIONS

We propose LogicGuard, a novel Temporal Logic-based critic, which augments and boosts the performance of existing off-the-shelf LLM planners by protecting them against unsafe and inefficient actions. Composing LogicGuard with an LLM planner leads to a novel symbolic actor-critic architecture designed to solve long-horizon planning tasks in dynamic, embodied environments using large language models (LLMs). This architecture breaks sequential planning into two timescales; an online actor proposes high-level actions based on current state descriptions, and an offline critic loop that imposes symbolic performance and safety constraints learned from past trajectories. This modular decomposition leverages LLMs' strengths in local reasoning and addresses their weaknesses in long-term consistency.

LogicGuard is domain-agnostic, naturally integrates with existing LLM-based planners, and is the first to use symbolic temporal logic as a communication protocol between the actor and critic, enabling interpretable, verifiable, and generalizable decision making. Our contributions are threefold.

1. **Symbolic actor-critic architecture**: We propose a two-timescale architecture where an LLM actor generates high-level actions online, and an LLM critic periodically analyzes trajectories offline to induce Linear Temporal Logic (LTL) constraints on the actor. These constraints prune unsafe or inefficient decisions, improving task performance, and are enforced online using an LTL verifier. Each constraint is accompanied by a natural language explanation, which is communicated back to the actor upon violation to guide subsequent decisions. Our architecture integrates LLM planning heuristics with formal performance guarantees, combining flexibility with reliable behavior.

2. **Communication via temporal logic**: We introduce a novel mechanism for actor-critic interaction based on symbolic temporal logic. In contrast to traditional reinforcement learning critics or natural language feedback, our critic outputs verifiable, machine-checkable LTL constraints. LTL constraints provide strong safety guarantees, enforces logical consistency, and enables constraint reuse across similar states and tasks. We also present a graph-theoretic abstraction of planning under temporal logic, which we use to constrain the critic, and provide constraints grounded in data, while reducing planning complexity.

3. **Empirical validation in generalist and specialist embodied settings:** We demonstrate gains in (i) completion rates on 100 short-horizon household tasks from the Behavior dataset, where atomic propositions are automatically derived across diverse environments, and (ii) efficiency and consistency in the long-horizon Minecraft diamond-mining task, where hand-engineered propositions capture domain structure. This demonstrates that our architecture benefits both generalist agents operating in varied environments and specialist agents in structured long-horizon domains.

Overall, LogicGuard advances the goal of robust, safe, and interpretable LLM-based decision making in complex, dynamic environments. It transforms temporal logic from a static filter to a closed-loop supervisory signal for LLM planners, providing verifiable safety, data-grounded performance improvements, and robustness to hallucinations. This structure yields consistent long-horizon behavior and transfers across distinct embodied domains without modifying the underlying LLM.

## 2 BACKGROUND AND PROBLEM FORMULATION

A key challenge in deploying LLM-based agents in embodied environments is their difficulty with coherent long-horizon planning. Without explicit goal formalization or structured guidance, they often act reactively or inconsistently, leading to unsafe and inefficient behavior that undermines reliability in human-agent teams (Fan et al., 2008). For real-world collaboration, agents must not only avoid unsafe actions but also demonstrate efficiency, competence, and interpretable reasoning.

We address this by developing a mechanism that provides formal guarantees on agent behavior while allowing for self-correction and improvement from structured feedback. Our framework uses symbolic constraints learned over time to guide LLM decision-making, ensuring safe and efficient behavior with limited data. We evaluate this approach across two settings: (i) short-horizon household tasks in the Behavior dataset, and (ii) the long-horizon challenge of mining a diamond in Minecraft.

## 2.1 PROBLEM STATEMENT

We formalize the problem of safe and efficient LLM planning in embodied environments. Consider an agent operating in a state space $\mathcal{S}$ and a finite high-level action space $\mathcal{A}_{\text{abstract}}$, where $|\mathcal{A}_{\text{abstract}}| = m$. The agent observes a representation of the state, which is modeled by $\phi_{\text{full}} \colon \mathcal{S} \to \mathcal{S}_{\text{full}}$. Unlike standard RL, we assume no explicit reward function, and instead rely on a natural language goal description (e.g., "make an iron pickaxe" or "obtain a diamond").

To enable formal reasoning, we introduce an abstract Boolean representation $\phi_{\text{abstract}} \colon \mathcal{S}_{\text{full}} \to \mathcal{S}_{\text{abstract}}$, where $\mathcal{S}_{\text{abstract}} \subseteq \{0,1\}^n$, encodes $n$ atomic propositions capturing salient environment features. Within $\mathcal{S}_{\text{abstract}}$, we define $\mathcal{S}_{\text{goal}}$, the set of states that satisfy task objectives, and $\mathcal{S}_{\text{unsafe}}$, the set of states that violate safety requirements. Agents may be regulated through two types of LTL constraints. Safety constraints are expert-authored rules that forbid transitions into $\mathcal{S}_{\text{unsafe}}$, such as collision avoidance. Performance constraints, in contrast, are automatically induced by LogicGuard from observed trajectories, eliminating redundant or suboptimal action patterns. Assuming all high-level actions have equal cost, the agent's objective is to minimize the number of primitive actions required to reach $\mathcal{S}_{\text{goal}}$ while satisfying all safety constraints with high probability.

## 2.2 LINEAR TEMPORAL LOGIC PRIMER

Linear Temporal Logic(LTL) (Pnueli, 1977) provides a formal language for expressing temporal properties over sequences of states. LTL formulas consist of variables called atomic propositions, Boolean operators, and temporal operators(discussed in Appendix A.1). LTL formulas can be converted into Büchi automata, which are finite-state machines enabling the algorithmic verification of whether a trajectory satisfies a given temporal specification. Using atomic propositions corresponding to physically meaningful events leads to interpretability. We employ SPOT (Duret-Lutz & Poitrenaud, 2004) to enforce LTL laws on the LLM actor. In our work, we allow the safety constraints, defining $\mathcal{S}_{\text{unsafe}}$ to have any user specified form. However, to control the LLM critic, we restrict it to a reactive fragment of LTL, only allowing it to impose constraints on the immediate next timestep.

## 2.3 EXPERIMENTAL DOMAINS

We evaluate LogicGuard in two contrasting embodied environments. Behavior (Li et al., 2023) consists of short-horizon household tasks in diverse environments. In contrast, Minecraft presents long-horizon compositional tasks with interdependent subgoals; we study the diamond-mining task (Guss et al., 2019). These domains allow us to evaluate LogicGuard on both generalist agents in diverse, short-horizon tasks and specialist agents in structured, long-horizon tasks.

## 3 METHODS

### 3.1 ARCHITECTURE OVERVIEW

We propose a hierarchical planning architecture in Figure 1 that integrates large language models (LLMs) with formal symbolic reasoning to achieve safe and efficient decision-making in complex environments. The architecture consists of two interacting loops operating at different timescales:

**Online actor loop:** At each timestep, an LLM actor receives a natural language state description, $x_{\text{full}} = \phi_{\text{full}}(s) \in \mathcal{S}_{\text{full}}$ and chooses a high-level action $a \in \mathcal{A}_{\text{abstract}}$. The action is checked against current LTL constraints using an LTL verifier. If valid, it is executed via a low level controller. Otherwise, the actor is informed of the violation with a natural language explanation and prompted to choose a new action. Examples of the LTL verifier in action are shown in Figure 2.

**Offline critic loop:** Periodically, an LLM-based critic analyzes completed trajectories to identify incorrect, inefficient or unsafe behaviors, proposing new LTL constraints or removing existing ones. Updates are immediately incorporated into the verifier, affecting subsequent actions.

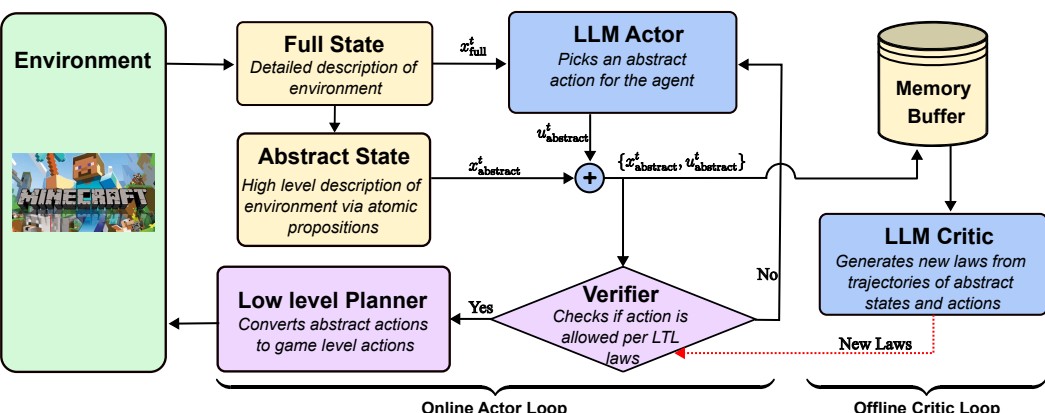

Figure 1: **LTL-guided actor-critic architecture:** Online, the LLM actor selects a high-level action, which a symbolic verifier checks against LTL safety and efficiency constraints. Valid actions are executed, while invalid actions are blocked and paired with a natural language explanation returned to the actor. Offline, LogicGuard reviews trajectories and proposes new LTL constraints to improve long-term behavior.

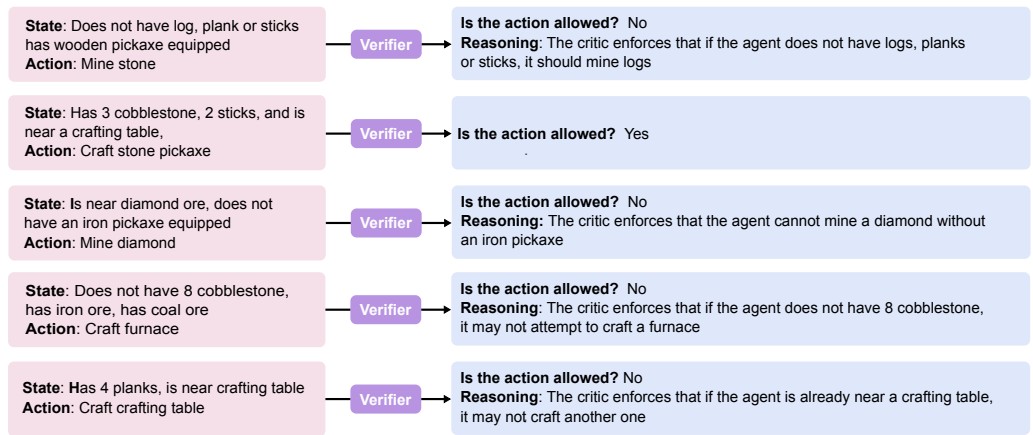

Figure 2: **Examples of the operation of the LTL-based verifier in a Minecraft Environment:** Each abstract state-action pair is checked against a Büchi automaton encoding existing LTL constraints. If the action violates any constraint, the verifier provides feedback identifying the violated rule, and the actor is prompted to replan.

The separation of online reactive planning and offline symbolic refinement enables safe, efficient and interpretable decision-making. The modular design also supports easy transfer across domains. A full architectural diagram is presented in Figure 1.

## 3.2 DEFINING THE ATOMIC PROPOSITIONS

The choice of variables in $\mathcal{S}_{\text{abstract}}$ directly affects the critic's expressive power and tractability. For generalist environments like Behavior, we automate the design of $\mathcal{S}_{\text{abstract}}$, initializing it with variables needed for goal satisfaction and safety constraints, and then augmenting it based on observed state changes during exploratory rollouts. This ensures the abstract state is both task-aware and environment-adaptive while avoiding excessive manual design (details in Appendix A.4.1).

### 3.3 PLANNING AS A GRAPH TRAVERSAL PROBLEM

We frame efficient planning as a shortest path problem under safety constraints. Each primitive action has unit cost, as LLM calls dominate execution time. The agent aims to reach a state in $\mathcal{S}_{\text{goal}}$ from its current state in as few steps as possible, while avoiding $\mathcal{S}_{\text{unsafe}}$.

We model the problem as a bipartite graph $\mathcal{G}$. One partition of this graph consists of symbolic states $\mathcal{S}_{\text{abstract}}$, while the other partition consists of $\mathcal{S}_{\text{abstract}} \times \mathcal{A}_{\text{abstract}}$. Edges from $s \in \mathcal{S}_{\text{abstract}}$ to $(s, a)$ exist only if $a$ is allowed by current LTL constraints; edges from $(s, a)$ to $s'$ represent state transitions. This bipartite structure explicitly separates the roles of the actor, which selects edges from states to state-action pairs, and the critic, which prunes edges via constraints.

Due to an exponential number of nodes, finding the shortest path in this graph is exponentially hard, motivating the need for LLMs to guide exploration via natural language reasoning, efficiently navigating the graph despite its combinatorial complexity.

### 3.4 LLM ACTOR: GUIDED EXPLORATION

The LLM actor receives a full state description $x_{\text{full}}$ and proposes a high-level action from the action space $\mathcal{A}$ such as "mine_stone" or "grasp_plywood". In the bipartite graph $\mathcal{G}$, the actor chooses an edge flowing out from the agents current state $x_{\text{abstract}}$, which is legal as per the current LTL constraints. The use of the LLM allows us to replace brute force exploration with semantically guided traversal in a large symbolic space. Our architecture allows the direct use of existing LLM planners. We adopt InnerMonologue (Huang et al., 2022b) for Behavior, and both SayCan (Ahn et al., 2022) and InnerMonologue for Minecraft. To ensure adaptivity and prevent overly restrictive rules, the actor tracks repeated attempts of violations of LTL constraints; if a particular rule is triggered beyond a fixed threshold, it is removed, allowing the agent to explore alternative strategies.

### 3.5 LOGICGUARD: THE LINEAR TEMPORAL LOGIC-BASED LLM CRITIC

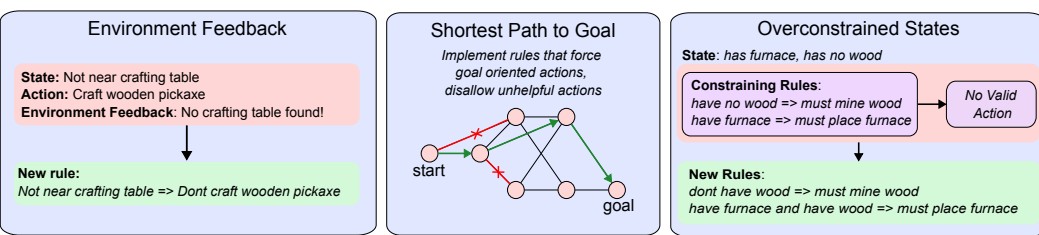

Figure 3: **Sources of LogicGuard generated laws:** LogicGuard generates new LTL laws by observing complete trajectories. Particularly, it generates laws based on three sources: environment feedback, graph-based efficiency improvements, and contradiction detection in over-constrained states.

The use of an LLM critic that communicates using LTL laws allows us to convert LTL from a static safety filter into a dynamic learning signal for performance enhancement in embodied LLM agents. In our experiments, we choose to run the critic to analyze complete trajectories. In practice the critic may be run more or less often depending on the requirements of the specific application. The critic is prompted to identify inefficient behaviors and to propose new reactive LTL constraints of the form

$$G(\phi_s \implies X(\phi_a)), \tag{1}$$

where $\phi_s$ is a boolean expression over symbolic state features, and $\phi_a$ specifies allowed or disallowed actions. These constraints eliminate inefficient behaviors. For example,

$$G(\text{agent\_has\_wooden\_pickaxe} \implies X(!\text{action\_craft\_wooden\_pickaxe})), \tag{2}$$

prevents crafting duplicate wooden pickaxes. Constraints are only generated if $\phi_s$ is observed in the trajectory, ensuring generalization grounded in data. Constraints are induced from three sources as shown in Figure 3:

1. **Environment feedback**: Actions deemed invalid by the environment (e.g. soaking a rag without turning on the tap or mining diamonds without an iron pickaxe) lead to an error message. Such actions are encoded into constraints to prevent future errors.

2. **Graph-based efficiency**: The critic is prompted to analyze trajectories in the context of the symbolic task graph from 3.3, classifying actions as efficient or inefficient, pruning wasteful actions, with an emphasis on repetitive actions. These laws are meant to detect and eliminate failure modes and repetitive actions in the actor's trajectory.

3. **Overconstrained States**: When current laws collectively eliminate all feasible actions in a state, the system falls back to bare minimum hand-engineered laws. Offline, these states are analyzed to refine or relax constraints, preventing overly restrictive rules.

Enforcing that the critic is only allowed to induce constraints grounded in trajectory data is very effective at preventing hallucinations. This natural choice yields several important advantages. First, all proposed rules are traceable. Second, we may bound the number of possible rules that are generated (See Remark 1). Third, the effect of each law on the actor's policy can be clearly quantified (See Remark 2). This quantification may be used to bound both potential improvement from iteration as well as potential harm from hallucination of the LLM.

**Remark 1.** *Let $(s_1, a_1, \ldots, s_N, a_N)$ be a trajectory with $s_i \in \mathcal{S}_{abstract}$, and each $a_i \in \{0, 1\}^m$ a one-hot vector representing an action from a finite action space $\mathcal{A}_{abstract}$ of size $m$. Consider an algorithm that generates LTL constraints of the form equation 1 where $\phi_s$ is a boolean condition over the symbolic state that holds for at least one $s_i$ in the trajectory and $\phi_a \in \{a_i, !a_i\}$.*

*Then, the algorithm can generate at most $N$ distinct such laws from the trajectory while ensuring that, for every state $s \in \mathcal{S}_{abstract}$, there exists at least one action $a \in \{0, 1\}^m$ satisfying all LTL laws.*

We provide the proof for this statement in Appendix A.2.

In our bipartite graph representation, pruning edges based on observed trajectories reduces complexity from $O(m \cdot 2^n)$ to linear in dataset size. Interpretable atomic propositions allow the critic to generalize constraints to semantically equivalent but unseen states. Finally, the sparse structure of goal-directed task graphs allows LogicGuard produces sample-efficient and robust behavior while enforcing both safety and efficiency.

**Remark 2.** *Let $\pi \colon \mathcal{S}_{abstract} \to \Delta(\mathcal{A}_{abstract})$ be a stochastic policy over a finite action space. For an action $a$ in state $s$, if $\pi(a|s) = p$, a reactive LTL law of the form equation 1 that blocks $a$ yields a policy $\pi_{block}$ such that $D_{\mathrm{KL}}(\pi_{block}(\cdot|s)||\pi(\cdot|s)) = -\log(1 - p)$ and an LTL law enforcing $a$ yields a policy $\pi_{force}$ such that $D_{\mathrm{KL}}(\pi_{force}(\cdot|s)||\pi(\cdot|s)) = -\log(p)$. Thus, each law changes the policy by a precisely quantifiable amount.*

We provide the proof for this statement in Appendix A.2

## 4 EXPERIMENTS

### 4.1 EXPERIMENTAL SETUP

Our actors use OpenAI GPT-4.1 as a backbone LLM and our critics use o3-mini. All temperatures are set to 0.1 to reduce stochasticity while still allowing exploration. As discussed in Section 2.3, we evaluate LogicGuard in two embodied environments. Implementation details including prompts and APs are presented in the appendix.

**Behavior:**  We use the Behavior (Li et al., 2023) dataset, consisting of short-horizon (average horizon: 14.6) household tasks with multiple independent subtasks. High-level actions and observations are designed via the API and goal specifications from Li et al. (2024). Completion rate is the primary metric, as prior work shows non–chain-of-thought LLMs struggle in this domain. To support diverse tasks and environments, atomic propositions are automatically generated, enabling LogicGuard for general diverse settings. We adopt InnerMonologue as the LLM actor. Given the diversity of tasks and environments, implementing a reliable affordance function is challenging, which limits the applicability of SayCan in this setting.

**Minecraft:**  Minecraft provides a partially observable environment with compositional, interdependent subgoals. We study the diamond-mining task (Guss et al., 2019), a long-horizon setting (see Table 2) requiring a sequence of dependent subgoals without intermediate rewards. We interface via the Mineflayer API (PrismarineJS Team, 2025), which provides structured observations and

path-planning utilities, upon which we design atomic actions. Evaluation metrics in this domain are efficiency (number of high-level actions required to obtain a diamond) and safety (number of failed or illegal actions). Unlike in Behavior, here we design hand-engineered atomic propositions, highlighting LogicGuard's ability to augment specialist agents for complex compositional goals.

## 4.2 BASELINES

Our modular architecture allows us to augment off-the-shelf LLM planners with our symbolic critic. We focus on two representative planners:

**InnerMonologue** (Huang et al., 2022b): An LLM planner that interleaves natural language thoughts and code actions, incorporating feedback at each step. This provides implicit reflection and sequential planning. We use InnerMonologue in both Behavior and Minecraft.

**SayCan** (Ahn et al., 2022) A two-stage planner that filters feasible actions via affordances and ranks them for goal relevance. We only evaluate SayCan in Minecraft, since affordance functions do not scale to Behavior's large, diverse action space. In Minecraft, SayCan is combined with LogicGuard for LTL-based affordance filtering.

In our experiments, the actor and critic loops run iteratively to improve performance on each trajectory: the actor first generates a trajectory, the critic reviews it and induces new constraints, and the actor then produces the next trajectory under these updated constraints. In Behavior, InnerMonologue undergoes two rounds of criticism per trajectory. In Minecraft, InnerMonologue also receives two rounds of criticism, while SayCan requires four to achieve stable performance. While LogicGuard supports hand-engineered LTL constraints, all experiments use only critic-generated laws, except for the SayCan baseline in Minecraft, which relies on a hand-engineered affordance function.

## 4.3 RESULTS

### 4.3.1 BEHAVIOR BENCHMARK: TASK COMPLETION

Behavior (Li et al., 2023) consists of 100 short-horizon household tasks with multiple independent subtasks (e.g., pick up objects, place items, open containers). Since tasks are short-horizon, we focus on task completion rather than efficiency. We end all trajectories after 40 actions, or if the actor chooses to declare it is done.

Table 1: Task completion rates on Behavior-100.

| Method | Completed Tasks |
|---|---|
| InnerMonologue | 47% |
| **InnerMonologue + LogicGuard** | **72**% |

### 4.3.2 MINECRAFT: EFFICIENCY AND SAFETY

In Minecraft, the agent must mine a diamond from scratch, involving long-horizon dependencies across mining and crafting subgoals. We evaluate efficiency (number of primitive actions to reach key subgoals) and safety (number of failed or unsafe actions).

**Efficiency** Table 2 shows the average number of primitives required to reach each subgoal. We note a 23% increment in the performance of InnerMonologue in the diamond mining task. SayCan is very easily distracted by the abstract nature of high level actions such as "explore", and does not make meaningful progress on the task. Our architecture identifies these drawbacks and blocks exploration related actions till the right tools are available.

**Safety** We measure failed actions and the number of unsafe actions blocked by the critic. LogicGuard significantly reduces both failure rates and unsafe actions (Table 3). Failures after LogicGuard are due to path planning errors within Mineshafter's API.

Table 2: Average primitive actions per subgoal (success rates in parentheses, error bars are standard deviations). Lower is better.

| Method | Wood Tool | Stone tool | Iron Tool | Diamond |
|---|---|---|---|---|
| SC | N/A (0/5) | N/A (0/5) | N/A (0/5) | N/A (0/5) |
| **SC + LogicGuard** | **12.6 ± 1.9(5/5)** | **17.6 ± 2.2 (5/5)** | **37.8 ± 4.6 (5/5)** | **45.4 ± 7.44(5/5)** |
| IM | 12.2 ± 1.2 (5/5) | 18.2 ± 1.7 (5/5) | 43.25 ± 4.5(4/5) | 45.5 ± 4.3 (4/5) |
| **IM + LogicGuard** | **9.4 ± 0.8 (5/5)** | **14.4 ± 0.8 (5/5)** | **32.0 ± 2.8 (5/5)** | **35.8 ± 2.5 (5/5)** |

IM = InnerMonologue, SC = SayCan.

Table 3: Agent safety metrics. We report the number of failed actions and the number of unsafe actions blocked by the critic (if applicable). Lower is better.

| Method | Failed Actions | Critic-Blocked Unsafe Actions |
|---|---|---|
| InnerMonologue | 23% | N/A |
| InnerMonologue + LogicGuard | 4.5% | 15% |

Our results highlight that LogicGuard generalizes across task structure and horizon, helping LLM actors act more reliably, safely, and efficiently.

**Ablations** To isolate the effect of introducing LTL into our architecture, we construct two ablations that progressively approach an "LLM-as-a-judge" implementation, motivated by Li et al. (2025) and Khan et al. (2025). We keep the set of laws frozen, since the critic LLM could, in principle, be equally expressive in both LTL and natural language when the atomic propositions are hand-engineered. LTL enters our actor loop in two ways: (i) the action verifier is an LTL verifier, and (ii) the actor prompt includes the list of all LTL-permitted actions to reduce erroneous LLM decisions.

The first ablation replaces the LTL verifier with an LLM verifier, while the second removes both the verifier and the LTL-derived action bias, yielding a full "LLM-as-a-judge" implementation. These ablations highlight several advantages of LogicGuard over using LLMs without formal verification.

- LLMs misinterpret the laws in $12.5\%$ of cases with only the LLM verifier, and in $11.6\%$ of cases on the LLM as a judge setup. While safety remains comparable to LogicGuard in our experiments, efficiency-related laws are frequently misinterpreted, reducing task efficiency. Importantly, the algorithm treats all laws equivalently; in other tasks or environments, an LLM-as-a-judge could threaten safety.

- In our first ablation, the actor is biased toward LTL-approved actions, but misinterpretations by the LLM produce deadlocks and unstable behavior.

- LogicGuard exhibits low variance and reproducible behavior. Although the LLM-as-a-judge architecture achieves comparable mean performance, it shows relatively high variance, occasionally leading to catastrophic efficiency failures.

All ablations are conducted with an InnerMonologue actor on the Minecraft environment due to consistent baseline behavior across runs. Detailed statistical results and sample traces of misinterpretations are provided in Appendix A.3.

## 5 DISCUSSION

### 5.1 LOGICGUARD MITIGATES SYSTEMATIC LLM ACTOR FAILURES

Naive LLM actors often repeat actions after task completion or ignore environment feedback, leading to inefficiency and loops. In Behavior, an actor may repeatedly pick up or place already organized objects; in Minecraft, it can mine blocks beyond task requirements. These failures stem from limited reasoning and prompt overload. LogicGuard addresses these issues by detecting redundant or failed actions and blocking them based on task conditions. This enforces interpretable, verifiable constraints, breaking loops and improving both reliability and task efficiency.

## 5.2 LOGICGUARD IMPROVES OVER LLM-AS-A-JUDGE ARCHITECTURES

LLM-as-a-Judge architectures (Li et al., 2025) typically rely on LLMs, either fine-tuned or prompted, with no formal verification. This approach is computationally expensive and a bottleneck in real-time control. In contrast, LogicGuard leverages the LLM to design rules in a structured, machine-readable language, which can be enforced efficiently at runtime, avoiding repeated LLM calls and overhead. Further, LLM-as-a-Judge systems operate as stochastic black boxes with no formal guarantees, and misinterpretations can lead to catastrophic failures in both safety and efficiency. In comparison, LogicGuard ensures that every action satisfies both user specified an LLM generated constraints. Further, LLM generated constraints are verifiably grounded in data. These verification processes in place enable formal guarantees, as well as predictable and reproducible performance across experiments. Formal verification makes LogicGuard a safer, more reliable, and more efficient alternative to purely LLM-based judgment approaches.

## 5.3 ATOMIC PROPOSITIONS ARE A KEY DESIGN CHOICE

Atomic propositions (APs) define the variables the critic uses to construct LTL constraints, directly controlling rule expressivity. In the Behavior dataset, LogicGuard's primary failure mode arises from insufficiently expressive APs. For instance, our current automated AP generation treats each item in the environment as an independent entity, which complicates combinatorial tasks. Consider placing four plates into four boxes such that each box contains at least one plate. There are 24 feasible solutions. Encoding a single LTL formula that captures all feasible final states is highly complex for the critic, which must account for all possible permutations. By contrast, for specialist agents (Minecraft) we can afford to hand engineer APs to precisely capture task-relevant state features. This targeted design simplifies rule generation, reduces combinatorial complexity, and enables more reliable constraint synthesis.

## 5.4 GENERALIST VS SPECIALIST AGENTS

LogicGuard improves performance across diverse domains. In Behavior, the environment is unknown and contains rules the LLM cannot anticipate (e.g., items must be inside a sink before soaking), which the critic must discover iteratively, akin to human learning. In Minecraft, extensive textual documentation allows the LLM to succeed with minimal guidance. Evaluating both domains demonstrates that LogicGuard supports generalist agents in novel settings and specialist agents in structured, long-horizon tasks.

## 6 CONCLUSION AND FUTURE WORK

We introduced a modular actor–critic architecture in which an LLM critic generates LTL constraints over abstracted trajectory representations, and an online verifier enforces these constraints on an LLM actor's actions. The critic operates at a slower timescale and applies zero-shot reasoning to identify and correct unsafe or inefficient behavior in a few iterations. Our modular architecture enables us to use existing off-the-shelf LLM planners as actors. The use of LTL constraints guarantees shielding of the LLM from unsafe or inefficient behavior. By expressing constraints in LTL over human-readable atomic propositions, we gain a symbolic structure that is immediately enforceable and human verifiable. We demonstrate our approach in two contrasting embodied domains, achieving significant improvements over baseline off-the-shelf LLM agents.

Several directions remain for future work. First, a critic could leverage annotated expert trajectories to imitate optimal behavior. Some initial work in this direction is presented in Vazquez-Chanlatte et al. (2025) and Gupta et al. (2024). Second, in dynamic environments, incorporating mathematical models of the environment into the critic could enable generation of LTL laws that evolve over time. Such adaptive laws are particularly relevant for multi-agent coordination and human-robot teaming, where modeling other agents can inform constraint synthesis. Finally, we aim to extend these ideas to real-world robotics, where online decisions may rely on smaller, potentially unreliable models that require formal constraints. Overall, our approach demonstrates that formal logic-based constraints provide a promising path toward safe, scalable, and general-purpose LLM agents.

## 7    ETHICS STATEMENT

Our work focuses on improving LLM-based planners in simulated environments (Behavior and Minecraft) and does not involve human subjects or sensitive data. LLMs can exhibit unsafe or unreliable behavior, and overreliance on them in real-world robotics could be hazardous. Our research hopes to draw attention and inspire further work towards safety nets for blackbox foundational models. Our framework is intended as a first step to improve reliability and safety in LLM-driven agents.

## 8    REPRODUCIBILITY STATEMENT

We provide detailed descriptions of all environments, prompts and LLM models used in our experiments. While we use OpenAI's GPT-4.1 and o3-mini APIs, which are inherently stochastic even at low temperature, we run several samples for each experiment in order to eliminate stochasticity.

### ACKNOWLEDGMENTS

We acknowledge the use of ChatGPT for assistance in improving the wording and grammar of this document. We thank the reviewers for pointing us towards literature on LLM as a judge and graph based planning in linear temporal logic.

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

## A    APPENDIX

### A.1    LINEAR TEMPORAL LOGIC OPERATORS

Linear Temporal Logic(LTL) (Pnueli, 1977) provides a formal language for expressing temporal properties over sequences of states. LTL formulas are built using the usual Boolean operators, with four temporal operators .

1. $X(\psi)$: $\psi$ holds at the next timestep.
2. $F(\psi)$: eventually $\psi$ holds.
3. $G(\psi)$: $\psi$ holds in all future states.
4. $\psi_1 U \psi_2$ means that $\psi_1$ holds until $\psi_2$ becomes true

### A.2    PROOFS OF REMARKS 1 AND 2

#### A.2.1    PROOF OF REMARK 1

There are at most $N$ unique values for $s_i$. Each constraint is of the form $G(\phi_s \Rightarrow X(\phi_a))$, and $\phi_s$ must hold for at least one $s_i$, the number of distinct $\phi_s$ that can be constructed from the trajectory is at most $N$. Further, the algorithm can only decide if each action is allowed or disallowed. As a consequence, the total number of laws is at most $2N$. However, since actions cannot be simultaneously allowed and disallowed, the number of actions an algorithm operating under our assumptions can make such that every state has at least one feasible action is at most $N$.

#### A.2.2    PROOF OF REMARK 2

If action $a^\star$ is blocked in state $s$, we assume that blocking a particular action sets its probability of occurring to 0, and re-normalizes the other probabilities. Since each probability of every other action scales by $\frac{1}{1-p}$

$$D_{\mathrm{KL}}(\pi_{block}(\cdot|s)||\pi(\cdot|s)) = \sum_{a \neq a^\star} \pi_{block}(a|s) \log \frac{\pi_{block}(a|s)}{\pi(a|s)} = \sum_{a \neq a^\star} \pi_{block}(a|s) \log \frac{1}{1-p}$$
$$= -\log(1-p)$$

Similarly, if action $a^\star$ is forced in state $s$, $\pi(a^\star|s) = 1$

$$D_{\mathrm{KL}}(\pi_{block}(\cdot|s)||\pi(\cdot|s)) = \pi_{force}(a^\star|s) \log \frac{\pi_{force}(a^\star|s)}{\pi(a^\star|s)} = -\log p$$

## A.3 DETAILED RESULTS OF ABLATIONS

### A.3.1 STATISTICAL RESULTS

We present the confusion matrix relating actions the LLM judge took vs the the actions an LTL verifier would have taken with the same set of laws. Note that due to no ambiguity in LTL, the LTL verifier's decision must be considered as ground truth. First, in Table 5 we present the disagreement confusion matrix between the LTL verifier and the LLM verifier in the LLM as a verifier setting. Next, the same confusion matrix is presented in Table 6 for the LLM-as-a-Judge setting. Note that the removal of the LTL law informed bias from the actor results in a larger number of rejected actions. Next, we summarize the performance of both methods in comparison to LogicGuard and no critic in Table 7. We note that while LLM-as-a-judge and LLM-as-a-verifier methods are only slightly worse than LogicGuard, LLM as a judge in particular has a high variance due to a catastrophic failure in efficiency due to a misinterpretation in one of the runs. Finally, we note that actions leading to environmental feedback are comparable across the ablations and LogicGuard in Table 8.

Table 4: Confusion matrices comparing LLM-as-Verifier and LLM-as-Judge ablations. Rows denote LTL decisions; columns denote LLM decisions.

Table 5: LLM-as-Verifier (264 samples)

| Decision Pair | LLM Blocked | LLM Allowed |
|---|---|---|
| **LTL Blocked** | 10.6% | 1.9% |
| **LTL Allowed** | 10.6% | 76.9% |

Table 6: LLM-as-a-Judge (326 samples)

| Decision Pair | LLM Blocked | LLM Allowed |
|---|---|---|
| **LTL Blocked** | 22.3% | 3.1% |
| **LTL Allowed** | 8.6% | 76.9% |

Table 7: Average primitive actions per subgoal (success rates in parentheses, error bars are standard deviations).

| Method | Wood Tool | Stone tool | Iron Tool | Diamond |
|---|---|---|---|---|
| IM | $12.2 \pm 1.2$ (5/5) | $18.2 \pm 1.7$ (5/5) | $43.25 \pm 4.5$(4/5) | $45.5 \pm 4.3$ (4/5) |
| IM + LLM judge | $10.4 \pm 0.8$ (5/5) | $15.6 \pm 1.0$ (5/5) | $41.2 \pm 13.4$(5/5) | $45.0 \pm 13.2$ (4/5) |
| IM + LLM Verifier | $10.4 \pm 0.9$ (5/5) | $16.0 \pm 0.9$ (5/5) | $37.8 \pm 3.8$(5/5) | $41.6 \pm 5.3$ (4/5) |
| **IM + LogicGuard** | $\mathbf{9.4 \pm 0.8}$ **(5/5)** | $\mathbf{14.4 \pm 0.8}$ **(5/5)** | $\mathbf{32.0 \pm 2.8}$ **(5/5)** | $\mathbf{35.8 \pm 2.5}$ **(5/5)** |

IM = InnerMonologue.

Table 8: Agent actions leading to environmental error

| Method | Failed Actions |
|---|---|
| InnerMonologue | 23% |
| InnerMonologue + LLM Verifier | 4.8% |
| InnerMonologue + LLM Judge | 3% |
| InnerMonologue + LogicGuard | 4.5% |

### A.3.2 Sample traces of misinterpretations

Using an LLM as a judge is not stable and prone to hallucinations. The judge LLM often contradicts itself. The following is an LLM output that commonly appeared when the agent attempted to craft a stone pickaxe. Note that the LLM acknowledges that no rule is broken, yet the feasibility is set to False. In practice, we remedied this by prompting the verifier to print the rules violated before making a decision on feasibility.

```
"feasible": 0,
"violated_rules": "You do not have enough cobblestone to craft a stone
    pickaxe. The rules require at least 3 cobblestone, 2 sticks, being
    near a crafting table, and not already having a stone pickaxe. While
    you have 3 cobblestone, 2 sticks, and are near a crafting table, you
    already have a wooden pickaxe equipped, but the rule 'If you dont
    have any of 3 cobblestone, 2 sticks, or arent near a crafting table,
    or if you already have a stone pickaxe, then dont craft a stone
    pickaxe.' applies. However, you do not have a stone pickaxe yet, so
    the only issue is that you have exactly 3 cobblestone, which is
    sufficient. Upon review, all requirements are met. Therefore, no
    rules are violated."
```

A primary cause of the efficiency gap between LogicGuard and LLM-as-a-judge architecture was the misinterpretation of the law that stopped the agent from acquiring raw iron if it had enough and forcing it to smelt the iron. However, the LLM prevented the agent from smelting any iron unless it had at least three raw iron. While this is not a safety critical example, something similar could possibly happen in a safety critical law in another environment.

```
    'await smeltItem(bot, "raw_iron", "oak_planks"); failed because the
        verifier requires that Rule: \'If you have three iron ore, are
        near a furnace, have fuel and do not have either of an iron
        pickaxe or 3 iron ingots, smelt iron.\' violated. The agent only
        has 2 raw_iron, not 3, and already has 2 iron ingots. Therefore,
        the preconditions for smelting iron are not met. \n', 'await
        smeltItem(bot, "raw_iron", "oak_planks"); failed because the
        verifier requires that Rule violated: To smelt iron, you must
        have three iron ore (raw_iron), be near a furnace, have fuel,
        and not already have either an iron pickaxe or 3 iron ingots.
        The agent only has 2 raw_iron and already has 2 iron ingots, so
        the preconditions are not met. \n', 'await mineBlock(bot,
        "iron_ore"); failed because the verifier requires that You do
        not have a wood pickaxe, stone pickaxe, or iron pickaxe
        equipped. According to the rules: \'If you dont have a wood
        pickaxe, a stone pickaxe, or an iron pickaxe equipped, do not
        mine iron ore.\' \n'
```

## A.4 Implementation details: Behavior

### A.4.1 Automation of atomic predicate generation

As discussed in our main paper, the choice of atomic predicates is crucial to the success of the critic. For the Behavior dataset we automate this process. First, we design a minimal set of APs to describe the goal state manually. Then, for the remaining APs, we use an API (Li et al., 2024) to detect actions and observations as per Table 9. If an action or an observation is detected, it is added to the AP dictionary. This way, the actor LLM filters an exponentially large space of APs.

| Source | Generated Atomic Propositions (APs) |
|---|---|
| Robot hands | If left/right hand holds an object, generate `<object>_in_hand`. |
| Object states | **For each object and state:**
**InsideRoomTypes**: `<obj>_in_<room>`
**Burnt**: `<obj>_is_burnt`
**Cooked**: `<obj>_is_cooked`
**Stained**: `<obj>_is_stained`
**Dusty**: `<obj>_is_dusty`
**Frozen**: `<obj>_is_frozen`
**HeatSourceOrSink**: `<obj>_is_heat_source_or_sink`
**Open**: `<obj>_is_open`
**Sliced**: `<obj>_is_sliced`
**Soaked**: `<obj>_is_soaked`
**ToggledOn**: `<obj>_is_toggled_on` |
| Object relations | **For each object pair $(o_1, o_2)$ (excluding floor/self):**
**Inside**: `<o1>_inside_<o2>`
**NextTo**: `<o1>_next_to_<o2>`
**OnFloor**: `<o1>_on_floor`
**OnTop**: `<o1>_on_top_of_<o2>`
**Under**: `<o1>_is_under_<o2>` |
| Actions (generic) | For each simple action (e.g., `open`, `close`, `slice`, `clean`), generate `<action>_<object>`. |
| Actions (binary) | For binary/2-argument actions (e.g., `place_ontop`, `place_inside`, `transfer_contents_ontop`, `place_nextto`), generate `<obj1>_<action>_<obj2>` for all ordered pairs with `obj1 != obj2`. |
| Termination | Always include the terminal proposition `done`. |

Table 9: Automatic generation rules for atomic propositions (APs) used in iGibson experiments. The code systematically maps robot inventory, per-object states, pairwise relations, and actions into sanitized propositional symbols for subsequent LTL processing.

## A.5 IMPLEMENTATION DETAILS: MINECRAFT

### A.5.1 MINEFLAYER API INTERFACE

We interface with Minecraft using the Mineflayer API, which exposes high-level observations as structured JSON objects and provides access to built-in path-planning routines. We define a set of action primitives on top of this interface to abstract the agent's decision space while preserving task complexity. Each primitive internally invokes Mineflayer's planners and lower-level control routines. The full list of primitives is as follows:

1. `mineBlock(bot, blockName)`: Mines 1 block of type `blockName`, provided it is visible within a 32-block radius.

2. `placeItem(bot, blockName, position)`: Places a block at a specified position, assuming the location is unoccupied and adjacent to an occupied block.

3. `craftItem(bot, itemName)`: Crafts 1 item of type `itemName`, assuming all ingredients are present in the agent's inventory. Recipes that require a crafting table assume one is nearby.

4. `smeltItem(bot, itemName, fuelName)`: Smelts 1 item of type `itemName` using `fuelName`, assuming both are present in the agent's inventory and a furnace is nearby.

5. `equipItem(bot, itemName, destination)`: Equips tools or armor. Some blocks require a minimum tool tier to mine, and the appropriate tool must be equipped.

6. `exploreUntil(bot, direction, condition)`: Causes the agent to explore in a specified direction until a user-defined condition is met (e.g., locating a specific block).

These primitives allow for rich compositional behavior while delegating locomotion to Mineflayer's planners. The full sequence of subgoals required to successfully mine a diamond in this setup is as follows:

1. Obtain wooden logs.

2. Craft logs into sticks and planks.

3. Craft a wooden pickaxe.

4. Mine stone blocks using the wooden pickaxe.

5. Craft a stone pickaxe and a furnace.

6. Obtain raw iron by mining iron blocks.

7. Smelt raw iron into iron ingots.

8. Craft an iron pickaxe.

9. Explore the world and mine a diamond block.

This structured task allows us to evaluate the ability of our actor-critic framework to perform multistage reasoning, tool use, and resource management over long horizons.

### A.5.2 LIST OF ATOMIC PROPOSITIONS IN MINECRAFT

We have two sets of atomic propositions, one for observations and one for high-level actions.

| Proposition | Meaning |
|---|---|
| obs_has_log | Agent has at least 1 log |
| obs_has_plank | Agent has at least 1 plank |
| obs_has_2x_plank | Agent has $\geq$ 2 planks |
| obs_has_3x_plank | Agent has $\geq$ 3 planks |
| obs_has_4x_plank | Agent has $\geq$ 4 planks |
| obs_has_11x_plank | Agent has $\geq$ 11 planks |
| obs_has_2x_stick | Agent has $\geq$ 2 sticks |
| obs_has_3x_cobble | Agent has $\geq$ 3 cobblestone |
| obs_has_8x_cobble | Agent has $\geq$ 8 cobblestone |
| obs_has_11x_cobble | Agent has $\geq$ 11 cobblestone |
| obs_has_wood_pickaxe | Agent has wooden pickaxe |
| obs_has_stone_pickaxe | Agent has stone pickaxe |
| obs_has_iron_pickaxe | Agent has iron pickaxe |
| obs_has_diamond | Agent has at least 1 diamond |
| obs_has_iron_ingot | Agent has $\geq$ 1 iron ingot |
| obs_has_3x_iron_ingot | Agent has $\geq$ 3 iron ingots |
| obs_has_1x_iron_ore | Agent has $\geq$ 1 iron ore |
| obs_has_2x_iron_ore | Agent has $\geq$ 2 iron ore |
| obs_has_3x_iron_ore | Agent has $\geq$ 3 iron ore |
| obs_has_crafting_table | Agent has crafting table |
| obs_has_furnace | Agent has furnace |
| obs_has_fuel | Agent has fuel (e.g., coal) |
| obs_near_crafting_table | Agent is near crafting table |
| obs_near_furnace | Agent is near furnace |
| obs_diamond_in_chunk | Diamonds detected nearby |
| obs_iron_in_chunk | Iron ore detected nearby |
| obs_coal_in_chunk | Coal detected nearby |
| obs_iron_pickaxe_equipped | Iron pickaxe is equipped |
| obs_stone_pickaxe_equipped | Stone pickaxe is equipped |
| obs_wood_pickaxe_equipped | Wooden pickaxe is equipped |

Table 10: Atomic propositions used for LTL constraints and their meanings.

| Action | Meaning |
|---|---|
| action_mine_log | Mine wood logs |
| action_mine_stone | Mine stone |
| action_mine_iron_ore | Mine iron ore |
| action_mine_coal | Mine coal |
| action_mine_diamond | Mine diamond |
| action_craft_planks | Craft planks from logs |
| action_craft_stick | Craft sticks from planks |
| action_craft_wooden_pickaxe | Craft wooden pickaxe |
| action_craft_stone_pickaxe | Craft stone pickaxe |
| action_craft_iron_pickaxe | Craft iron pickaxe |
| action_craft_crafting_table | Craft a crafting table |
| action_craft_furnace | Craft a furnace |
| action_smelt_iron | Smelt iron ore into ingots |
| action_equip_wood_pickaxe | Equip wooden pickaxe |
| action_equip_stone_pickaxe | Equip stone pickaxe |
| action_equip_iron_pickaxe | Equip iron pickaxe |
| action_explore_general | Explore randomly |
| action_explore_diamond_down | Explore downward for diamonds |
| action_place_crafting_table | Place crafting table |
| action_place_furnace | Place furnace |

Table 11: Action variables used in planning and their associated meanings.

### A.5.3 LIST OF LTL LAWS IMPOSED FOR EACH ACTOR

**SayCan**   The hard-hand engineered safety rules that prevent illegal actions are given below:

1. LTL: $G(\neg\mathsf{obs\_iron\_pickaxe\_equipped} \to X(\neg\mathsf{action\_mine\_diamond}))$
   Explanation: Diamonds cannot be mined unless an iron pickaxe is equipped.

2. LTL: $G(\neg\mathsf{obs\_near\_crafting\_table} \to X(\neg\mathsf{action\_craft\_wooden\_pickaxe} \wedge \neg\mathsf{action\_craft\_stone\_pickaxe} \wedge \neg\mathsf{action\_craft\_iron\_pickaxe}))$
   Explanation: Cannot craft any type of pickaxe unless near a crafting table.

3. LTL: $G(\neg\mathsf{obs\_near\_crafting\_table} \to X(\neg\mathsf{action\_craft\_furnace}))$
   Explanation: Cannot craft a furnace unless near a crafting table.

4. LTL: $G(\neg(\mathsf{obs\_stone\_pickaxe\_equipped} \vee \mathsf{obs\_iron\_pickaxe\_equipped}) \to X(\neg\mathsf{action\_mine\_iron\_ore}))$
   Explanation: Cannot mine iron ore unless a stone or iron pickaxe is equipped.

5. LTL: $G(\neg(\mathsf{obs\_wood\_pickaxe\_equipped} \vee \mathsf{obs\_stone\_pickaxe\_equipped} \vee \mathsf{obs\_iron\_pickaxe\_equipped}) \to X(\neg\mathsf{action\_mine\_stone}))$
   Explanation: Cannot mine stone unless any pickaxe is equipped.

6. LTL: $G(\neg\mathsf{obs\_has\_iron\_pickaxe} \to X(\neg\mathsf{action\_equip\_iron\_pickaxe}))$
   Explanation: Cannot equip an iron pickaxe unless the agent has one.

7. LTL: $G(\neg\mathsf{obs\_has\_3x\_plank} \vee \neg\mathsf{obs\_has\_2x\_stick} \to X(\neg\mathsf{action\_craft\_wooden\_pickaxe}))$
   Explanation: Cannot craft a wooden pickaxe without enough planks and sticks.

8. LTL: $G(\neg\mathsf{obs\_has\_4x\_plank} \to X(\neg\mathsf{action\_craft\_crafting\_table}))$
   Explanation: Cannot craft a crafting table without 4 planks.

9. LTL: $G(\neg\mathsf{obs\_has\_8x\_cobble} \to X(\neg\mathsf{action\_craft\_furnace}))$
   Explanation: Cannot craft a furnace without 8 cobblestone.

10. LTL: $G(\neg(\mathsf{obs\_has\_2x\_stick} \wedge \mathsf{obs\_has\_3x\_iron\_ingot}) \to X(\neg\mathsf{action\_craft\_iron\_pickaxe}))$
    Explanation: Cannot craft an iron pickaxe without 2 sticks and 3 iron ingots.

11. LTL: $G(\neg\mathsf{obs\_has\_3x\_cobble} \vee \neg\mathsf{obs\_has\_2x\_stick} \to X(\neg\mathsf{action\_craft\_stone\_pickaxe}))$
    Explanation: Cannot craft a stone pickaxe without cobblestone and sticks.

12. LTL: $G(\neg\mathsf{obs\_coal\_in\_chunk} \vee \neg(\mathsf{obs\_wood\_pickaxe\_equipped} \vee \mathsf{obs\_stone\_pickaxe\_equipped} \vee \mathsf{obs\_iron\_pickaxe\_equipped}) \to X(\neg\mathsf{action\_mine\_coal}))$
    Explanation: Cannot mine coal unless coal is nearby and a pickaxe is equipped.

13. LTL: $G(\neg\mathsf{obs\_has\_log} \to X(\neg\mathsf{action\_craft\_planks}))$
    Explanation: Cannot craft planks without logs.

14. LTL: $G(\neg\mathsf{obs\_has\_2x\_plank} \to X(\neg\mathsf{action\_craft\_stick}))$
    Explanation: Cannot craft sticks without 2 planks.

15. LTL: $G(\neg\mathsf{obs\_near\_furnace} \vee \neg\mathsf{obs\_has\_1x\_iron\_ore} \vee \neg\mathsf{obs\_has\_fuel} \to X(\neg\mathsf{action\_smelt\_iron}))$
    Explanation: Cannot smelt iron without a furnace, fuel, and iron ore.

16. LTL: $G(\neg\mathsf{obs\_has\_wood\_pickaxe} \to X(\neg\mathsf{action\_equip\_wood\_pickaxe}))$
    Explanation: Cannot equip a wooden pickaxe unless the agent has one.

17. LTL: $G(\neg\mathsf{obs\_has\_stone\_pickaxe} \to X(\neg\mathsf{action\_equip\_stone\_pickaxe}))$
    Explanation: Cannot equip a stone pickaxe unless the agent has one.

18. LTL: $G(\neg\mathsf{obs\_has\_crafting\_table} \to X(\neg\mathsf{action\_place\_crafting\_table}))$
    Explanation: Cannot place a crafting table unless the agent has one.

19. LTL: $G(\neg\mathsf{obs\_has\_furnace} \to X(\neg\mathsf{action\_place\_furnace}))$
    Explanation: Cannot place a furnace unless the agent has one.

The soft LTL rules implemented by the critic are as follows:

1. LTL: $G(\neg \text{obs\_has\_log} \wedge \neg \text{obs\_has\_plank} \wedge \neg \text{obs\_has\_2x\_stick} \wedge \neg \text{obs\_has\_iron\_pickaxe} \rightarrow X(\text{action\_mine\_log}))$
   Explanation: If the agent lacks logs, planks, sticks, and an iron pickaxe, it should mine wood logs.

2. LTL: $G(\text{obs\_has\_log} \wedge \neg \text{obs\_has\_plank} \rightarrow X(\text{action\_craft\_planks}))$
   Explanation: If the agent has logs but no planks, it should craft planks.

3. LTL: $G(\text{obs\_has\_2x\_plank} \wedge \neg \text{obs\_has\_2x\_stick} \rightarrow X(\text{action\_craft\_stick}))$
   Explanation: If the agent has planks but no sticks, it should craft sticks.

4. LTL: $G(\text{obs\_has\_4x\_plank} \wedge \text{obs\_has\_2x\_stick} \wedge \neg \text{obs\_has\_crafting\_table} \wedge \neg \text{obs\_near\_crafting\_table} \rightarrow X(\text{action\_craft\_crafting\_table}))$
   Explanation: If the agent has 4 planks and 2 sticks but no crafting table is nearby or already crafted, it must craft a crafting table.

5. LTL: $G(\text{obs\_has\_4x\_plank} \wedge \text{obs\_has\_2x\_stick} \wedge \text{obs\_has\_crafting\_table} \wedge \neg \text{obs\_near\_crafting\_table} \rightarrow X(\text{action\_place\_crafting\_table}))$
   Explanation: If the agent has 4 planks, 2 sticks, and a crafting table but is not near one, it should place the crafting table.

6. LTL: $G(\text{obs\_has\_3x\_plank} \wedge \text{obs\_has\_2x\_stick} \wedge \neg \text{obs\_has\_wood\_pickaxe} \wedge \text{obs\_near\_crafting\_table} \rightarrow X(\text{action\_craft\_wooden\_pickaxe}))$
   Explanation: If the agent has 3 planks, 2 sticks, is near a crafting table, and doesn't already have a wooden pickaxe, it should craft one.

7. LTL: $G(\text{obs\_has\_wooden\_pickaxe} \wedge \neg \text{obs\_wooden\_pickaxe\_equipped} \wedge \neg \text{obs\_has\_3x\_cobble} \rightarrow X(\text{action\_equip\_wooden\_pickaxe}))$
   Explanation: If the agent has a wooden pickaxe but it isn't equipped and it lacks three cobblestones, it should equip the pickaxe.

8. LTL: $G(\text{obs\_wood\_pickaxe\_equipped} \wedge \neg \text{obs\_has\_3x\_cobble} \wedge \neg \text{obs\_has\_stone\_pickaxe} \wedge \neg \text{obs\_has\_2x\_stick} \rightarrow X(\text{action\_mine\_stone}))$
   Explanation: If equipped with a wooden pickaxe and lacking stone, the agent should mine cobblestone.

9. LTL: $G(\text{obs\_wood\_pickaxe\_equipped} \wedge \text{obs\_has\_3x\_cobble} \wedge \neg \text{obs\_has\_stone\_pickaxe} \rightarrow X(\text{action\_craft\_stone\_pickaxe}))$
   Explanation: If the agent has 3 cobblestones and a wooden pickaxe equipped, it should craft a stone pickaxe.

10. LTL: $G(\text{obs\_has\_3x\_iron\_ore} \wedge \text{obs\_near\_furnace} \wedge \text{obs\_has\_fuel} \wedge \neg \text{obs\_has\_3x\_iron\_ingot} \rightarrow X(\text{action\_smelt\_iron}))$
    Explanation: If the agent has 3 iron ore, is near a furnace, has fuel, and lacks 3 ingots, it should smelt iron.

11. LTL: $G(\text{obs\_has\_3x\_iron\_ingot} \wedge \text{obs\_has\_2x\_stick} \wedge \text{obs\_near\_crafting\_table} \wedge \neg \text{obs\_has\_iron\_pickaxe} \rightarrow X(\text{action\_craft\_iron\_pickaxe}))$
    Explanation: If the agent has the ingredients but no iron pickaxe, it should craft one.

12. LTL: $G(\text{obs\_has\_iron\_pickaxe} \wedge \neg \text{obs\_iron\_pickaxe\_equipped} \wedge \neg \text{obs\_has\_2x\_stick} \rightarrow X(\text{action\_equip\_iron\_pickaxe}))$
    Explanation: If the agent owns an iron pickaxe but hasn't equipped it, it should equip the pickaxe.

13. LTL: $G(\text{obs\_iron\_pickaxe\_equipped} \wedge \neg \text{obs\_diamond\_in\_chunk} \rightarrow X(\text{action\_explore\_diamond\_down}))$
    Explanation: If an iron pickaxe is equipped and no diamonds are nearby, explore downward.

14. LTL: $G(\neg \text{obs\_has\_iron\_pickaxe} \wedge \neg \text{obs\_iron\_pickaxe\_equipped} \rightarrow X(\neg \text{action\_explore\_diamond\_down}))$
    Explanation: Do not explore for diamonds unless an iron pickaxe is available and equipped.

15. LTL: $G(\text{obs\_has\_fuel} \wedge \text{obs\_iron\_in\_chunk} \wedge \text{obs\_coal\_in\_chunk} \rightarrow X(\neg \text{action\_explore\_general}))$
    Explanation: Do not explore if iron and coal are already known to be nearby.

**InnerMonologue** Since our safety violations in InnerMonologue simply lead to environmental feedback and new laws, we do not include any hand-engineered safety laws in InnerMonologue.

- LTL: $G(\neg\mathsf{obs\_has\_log} \wedge \neg\mathsf{obs\_has\_plank} \wedge \neg\mathsf{obs\_has\_2x\_stick} \wedge \neg\mathsf{obs\_has\_iron\_pickaxe} \rightarrow X(\mathsf{action\_mine\_log}))$
  Explanation: If you don't have logs, planks, or sticks, mine logs.

- LTL: $G(\mathsf{obs\_has\_log} \wedge \neg\mathsf{obs\_has\_plank} \rightarrow X(\mathsf{action\_craft\_planks}))$
  Explanation: If you have logs but no planks, craft planks.

- LTL: $G(\mathsf{obs\_has\_4x\_plank} \wedge \neg\mathsf{obs\_near\_crafting\_table} \wedge \neg\mathsf{obs\_has\_crafting\_table} \rightarrow X(\mathsf{action\_craft\_crafting\_table}))$
  Explanation: If you have 4 planks, aren't near a crafting table, and don't have one, craft a crafting table.

- LTL: $G(\mathsf{obs\_has\_crafting\_table} \wedge \mathsf{obs\_has\_plank} \wedge \neg\mathsf{obs\_near\_crafting\_table} \rightarrow X(\mathsf{action\_place\_crafting\_table}))$
  Explanation: If you have a crafting table and a plank, but aren't near one, place the crafting table.

- LTL: $G(\mathsf{obs\_has\_3x\_plank} \wedge \mathsf{obs\_has\_2x\_stick} \wedge \mathsf{obs\_near\_crafting\_table} \wedge \neg\mathsf{obs\_has\_wood\_pickaxe} \rightarrow X(\mathsf{action\_craft\_wooden\_pickaxe}))$
  Explanation: If you have 3 planks, 2 sticks, are near a crafting table, and don't have a wooden pickaxe, craft one.

- LTL: $G(\mathsf{obs\_has\_3x\_cobble} \wedge \mathsf{obs\_has\_2x\_stick} \wedge \mathsf{obs\_near\_crafting\_table} \wedge \neg\mathsf{obs\_has\_stone\_pickaxe} \rightarrow X(\mathsf{action\_craft\_stone\_pickaxe}))$
  Explanation: If you have 3 cobble, 2 sticks, are near a crafting table, and don't have a stone pickaxe, craft one.

- LTL: $G(\mathsf{obs\_has\_8x\_cobble} \wedge \mathsf{obs\_near\_crafting\_table} \wedge \neg\mathsf{obs\_has\_furnace} \wedge \neg\mathsf{obs\_near\_furnace} \rightarrow X(\mathsf{action\_craft\_furnace}))$
  Explanation: If you have 8 cobble, are near a crafting table, and don't have or see a furnace, craft one.

- LTL: $G(\mathsf{obs\_has\_3x\_iron\_ore} \wedge \mathsf{obs\_near\_furnace} \wedge \mathsf{obs\_has\_fuel} \wedge \neg(\mathsf{obs\_has\_iron\_pickaxe} \vee \mathsf{obs\_has\_3x\_iron\_ingot}) \rightarrow X(\mathsf{action\_smelt\_iron}))$
  Explanation: If you have iron ore and fuel, are near a furnace, and don't already have iron ingots or a pickaxe, smelt iron.

- LTL: $G(\mathsf{obs\_has\_3x\_iron\_ingot} \wedge \mathsf{obs\_has\_2x\_stick} \wedge \mathsf{obs\_near\_crafting\_table} \wedge \neg\mathsf{obs\_has\_iron\_pickaxe} \rightarrow X(\mathsf{action\_craft\_iron\_pickaxe}))$
  Explanation: If you have 3 iron ingots, 2 sticks, are near a crafting table, and don't have an iron pickaxe, craft one.

- LTL: $G(\mathsf{obs\_has\_iron\_pickaxe} \wedge \neg\mathsf{obs\_iron\_pickaxe\_equipped} \rightarrow X(\mathsf{action\_equip\_iron\_pickaxe}))$
  Explanation: If you have an iron pickaxe but it's not equipped, equip it.

- LTL: $G(\mathsf{obs\_diamond\_in\_chunk} \wedge \mathsf{obs\_iron\_pickaxe\_equipped} \rightarrow X(\mathsf{action\_mine\_diamond}))$
  Explanation: If diamonds are nearby and an iron pickaxe is equipped, mine the diamond.

- LTL: $G(\neg\mathsf{obs\_diamond\_in\_chunk} \wedge \mathsf{obs\_iron\_pickaxe\_equipped} \rightarrow X(\mathsf{action\_explore\_diamond\_down}))$
  Explanation: If no diamonds are visible and an iron pickaxe is equipped, explore downward for diamonds.

- LTL: $G(\mathsf{obs\_diamond\_in\_chunk} \vee \neg\mathsf{obs\_iron\_pickaxe\_equipped} \rightarrow X(\neg\mathsf{action\_explore\_diamond\_down}))$
  Explanation: If diamonds are visible or no iron pickaxe is equipped, do not explore downward.

- LTL: $G(\neg\mathsf{obs\_wood\_pickaxe\_equipped} \wedge \neg\mathsf{obs\_stone\_pickaxe\_equipped} \wedge \neg\mathsf{obs\_iron\_pickaxe\_equipped} \rightarrow X(\neg\mathsf{action\_mine\_stone}))$
  Explanation: If no pickaxe is equipped, don't mine stone.

- LTL: $G(\neg\textsf{obs\_stone\_pickaxe\_equipped} \ \wedge \ \neg\textsf{obs\_iron\_pickaxe\_equipped} \ \rightarrow \ X(\neg\textsf{action\_mine\_iron\_ore}))$
  `Explanation:` Don't mine iron ore unless a stone or iron pickaxe is equipped.

- LTL: $G(\neg\textsf{obs\_has\_8x\_cobble} \rightarrow X(\neg\textsf{action\_craft\_furnace}))$
  `Explanation:` Don't craft a furnace without 8 cobblestone.

- LTL: $G(\neg\textsf{obs\_wood\_pickaxe\_equipped} \ \wedge \ \neg\textsf{obs\_stone\_pickaxe\_equipped} \ \wedge \ \neg\textsf{obs\_iron\_pickaxe\_equipped} \rightarrow X(\neg\textsf{action\_mine\_coal}))$
  `Explanation:` Don't mine coal without a pickaxe equipped.

- LTL: $G(\neg\textsf{obs\_has\_3x\_plank} \vee \neg\textsf{obs\_has\_2x\_stick} \vee \neg\textsf{obs\_near\_crafting\_table} \vee \textsf{obs\_has\_wood\_pickaxe} \rightarrow X(\neg\textsf{action\_craft\_wooden\_pickaxe}))$
  `Explanation:` Don't craft a wooden pickaxe unless you have the materials and don't already have one.

- LTL: $G(\neg\textsf{obs\_has\_3x\_cobble} \vee \neg\textsf{obs\_has\_2x\_stick} \vee \neg\textsf{obs\_near\_crafting\_table} \vee \textsf{obs\_has\_stone\_pickaxe} \rightarrow X(\neg\textsf{action\_craft\_stone\_pickaxe}))$
  `Explanation:` Don't craft a stone pickaxe unless you have materials and don't already have one.

- LTL: $G(\neg\textsf{obs\_has\_3x\_iron\_ingot} \vee \neg\textsf{obs\_has\_2x\_stick} \vee \neg\textsf{obs\_near\_crafting\_table} \vee \textsf{obs\_has\_iron\_pickaxe} \rightarrow X(\neg\textsf{action\_craft\_iron\_pickaxe}))$
  `Explanation:` Don't craft an iron pickaxe unless you have materials and don't already have one.

## A.6 PROMPTS

Here, we provide the general prompts that guide the LLM. Most prompts are implemented as Python f-strings; to keep them concise, we show the templates without substituting the variable names.

### A.6.1 BEHAVIOR

**Actor prompts** First, the context prompt:

```
1  Problem:
2  You are designing instructions for a household robot.
3  The goal is to guide the robot to modify its environment from its
       current state to a desired final state.
4  The input will be the current environment state, the target environment
       state, the objects you can interact with in the environment.
5  The output should be the next action command that the robot may execute
       in order to make progress towards achieving the target state.
6
7  Data format: After # is the explanation.
8
9  Format of the states:
10 The current environment state is described as a list of dictionaries.
       Each dictionary describes an object, its category, followed by its
       description, which includes several of its properties including a
       description of its location.
11 For example:
12 {'name': 'plywood_1',
13 'category': 'plywood',
14 'State description ':
15 ['Location: living_room', 'Stain status: Clean', 'Dust status: Clean',
       'Touching: room_floor_living_room_0', 'Touching: plywood_0',
       'Touching: room_floor_kitchen_0', 'On top of:
       room_floor_living_room_0', 'On top of: room_floor_kitchen_0', 'On
       floor: room_floor_living_room_0', 'Next to: plywood_0']}
16
17 You will be provided with the environment state of each object in the
       environment in the above format.
18
19 Format of the action commands:
```

```
Action commands is a dictionary with the following format:
{
        \"action\": \"action_name\",
        \"object\": \"target_obj_name\",
        \"thoughts\": \"inner monologue describing why this action is
            chosen\",
}

or

{
        \"action\": \"action_name\",
        \"object\": \"target_obj_name1,target_obj_name2\",
        \"thoughts\": \"inner monologue describing why this action is
            chosen\",
}

The action_name must be one of the following:
LEFT_GRASP # the robot grasps the object with its left hand, to execute
    the action, the robot's left hand must be empty, e.g. {'action':
    'LEFT_GRASP', 'object': 'apple_0'}.
RIGHT_GRASP # the robot grasps the object with its right hand, to
    execute the action, the robot's right hand must be empty, e.g.
    {'action': 'RIGHT_GRASP', 'object': 'apple_0'}.
LEFT_PLACE_ONTOP # the robot places the object in its left hand on top
    of the target object and release the object in its left hand, e.g.
    {'action': 'LEFT_PLACE_ONTOP', 'object': 'table_1'}.
RIGHT_PLACE_ONTOP # the robot places the object in its right hand on top
    of the target object and release the object in its left hand, e.g.
    {'action': 'RIGHT_PLACE_ONTOP', 'object': 'table_1'}.
LEFT_PLACE_INSIDE # the robot places the object in its left hand inside
    the target object and release the object in its left hand, to
    execute the action, the robot's left hand must hold an object, and
    the target object can't be closed e.g. {'action':
    'LEFT_PLACE_INSIDE', 'object': 'fridge_1'}.
RIGHT_PLACE_INSIDE # the robot places the object in its right hand
    inside the target object and release the object in its left hand, to
    execute the action, the robot's right hand must hold an object, and
    the target object can't be closed, e.g. {'action':
    'RIGHT_PLACE_INSIDE', 'object': 'fridge_1'}.
RIGHT_RELEASE # the robot directly releases the object in its right
    hand, to execute the action, the robot's left hand must hold an
    object, e.g. {'action': 'RIGHT_RELEASE', 'object': 'apple_0'}.
LEFT_RELEASE # the robot directly releases the object in its left hand,
    to execute the action, the robot's right hand must hold an object,
    e.g. {'action': 'LEFT_RELEASE', 'object': 'apple_0'}.
OPEN # the robot opens the target object, to execute the action, the
    target object should be openable and closed, also, toggle off the
    target object first if want to open it, e.g. {'action': 'OPEN',
    'object': 'fridge_1'}.
CLOSE # the robot closes the target object, to execute the action, the
    target object should be openable and open, e.g. {'action': 'CLOSE',
    'object': 'fridge_1'}.
COOK # the robot cooks the target object, to execute the action, the
    target object should be put in a pan, e.g. {'action': 'COOK',
    'object': 'apple_0'}.
CLEAN # the robot cleans the target object, to execute the action, the
    robot should have a cleaning tool such as rag, the cleaning tool
    should be soaked if possible, or the target object should be put
    into a toggled on cleaner like a sink or a dishwasher, e.g.
    {'action': 'CLEAN', 'object': 'window_0'}.
FREEZE # the robot freezes the target object e.g. {'action': 'FREEZE',
    'object': 'apple_0'}.
UNFREEZE # the robot unfreezes the target object, e.g. {'action':
    'UNFREEZE', 'object': 'apple_0'}.
```

```
50  SLICE # the robot slices the target object, to execute the action, the
        robot should have a knife in hand, e.g. {'action': 'SLICE',
        'object': 'apple_0'}.
51  SOAK # the robot soaks the target object, to execute the action, the
        target object must be put in a toggled on sink, e.g. {'action':
        'SOAK', 'object': 'rag_0'}.
52  DRY # the robot dries the target object, e.g. {'action': 'DRY',
        'object': 'rag_0'}.
53  TOGGLE_ON # the robot toggles on the target object, to execute the
        action, the target object must be closed if the target object is
        openable and open e.g. {'action': 'TOGGLE_ON', 'object': 'light_0'}.
54  TOGGLE_OFF # the robot toggles off the target object, e.g. {'action':
        'TOGGLE_OFF', 'object': 'light_0'}.
55  LEFT_PLACE_NEXTTO # the robot places the object in its left hand next to
        the target object and release the object in its left hand, e.g.
        {'action': 'LEFT_PLACE_NEXTTO', 'object': 'table_1'}.
56  RIGHT_PLACE_NEXTTO # the robot places the object in its right hand next
        to the target object and release the object in its right hand, e.g.
        {'action': 'RIGHT_PLACE_NEXTTO', 'object': 'table_1'}.
57  LEFT_TRANSFER_CONTENTS_INSIDE # the robot transfers the contents in the
        object in its left hand inside the target object, e.g. {'action':
        'LEFT_TRANSFER_CONTENTS_INSIDE', 'object': 'bow_1'}.
58  RIGHT_TRANSFER_CONTENTS_INSIDE # the robot transfers the contents in the
        object in its right hand inside the target object, e.g. {'action':
        'RIGHT_TRANSFER_CONTENTS_INSIDE', 'object': 'bow_1'}.
59  LEFT_TRANSFER_CONTENTS_ONTOP # the robot transfers the contents in the
        object in its left hand on top of the target object, e.g. {'action':
        'LEFT_TRANSFER_CONTENTS_ONTOP', 'object': 'table_1'}.
60  RIGHT_TRANSFER_CONTENTS_ONTOP # the robot transfers the contents in the
        object in its right hand on top of the target object, e.g.
        {'action': 'RIGHT_TRANSFER_CONTENTS_ONTOP', 'object': 'table_1'}.
61  LEFT_PLACE_NEXTTO_ONTOP # the robot places the object in its left hand
        next to target object 1 and on top of the target object 2 and
        release the object in its left hand, e.g. {'action':
        'LEFT_PLACE_NEXTTO_ONTOP', 'object': 'window_0, table_1'}.
62  RIGHT_PLACE_NEXTTO_ONTOP # the robot places the object in its right hand
        next to object 1 and on top of the target object 2 and release the
        object in its right hand, e.g. {'action':
        'RIGHT_PLACE_NEXTTO_ONTOP', 'object': 'window_0, table_1'}.
63  LEFT_PLACE_UNDER # the robot places the object in its left hand under
        the target object and release the object in its left hand, e.g.
        {'action': 'LEFT_PLACE_UNDER', 'object': 'table_1'}.
64  RIGHT_PLACE_UNDER # the robot places the object in its right hand under
        the target object and release the object in its right hand, e.g.
        {'action': 'RIGHT_PLACE_UNDER', 'object': 'table_1'}.
65  DONE # the robot has achieved the target environment as per your best
        judgement, e.g. {'action': 'DONE', 'object': 'none'}.
66
67  Format of the interactable objects:
68  Interactable object will contain multiple lines, each line is a
        dictionary with the following format:
69  {
70      \"name\": \"object_name\",
71      \"category\": \"object_category\"
72  }
73  object_name is the name of the object, which you must use in the action
        command, object_category is the category of the object, which
        provides a hint for you in interpreting initial and goal condtions.
74
75
76  thoughts: This is your inner monologue describing why you choose this
        action, it will be used as a feedback to improve your next action
        command.
77
78  Please pay special attention:
```

```
79  1. The robot can only hold one object in each hand.
80  2. Action name must be one of the above action names, and the object
       name must be one of the object names listed in the interactable
       objects.
81  3. All PLACE actions will release the object in the robot's hand, you
       don't need to explicitly RELEASE the object after the PLACE action.
82  4. For LEFT_PLACE_NEXTTO_ONTOP and RIGHT_PLACE_NEXTTO_ONTOP, the action
       command are in the format of {'action': 'action_name', 'object':
       'obj_name1, obj_name2'}
83  5. If you want to perform an action to an target object, you must make
       sure the target object is not inside a closed object.
84  6. For actions like OPEN, CLOSE, SLICE, COOK, CLEAN, SOAK, DRY, FREEZE,
       UNFREEZE, TOGGLE_ON, TOGGLE_OFF, at least one of the robot's hands
       must be empty, and the target object must have the corresponding
       property like they're openable, toggleable, etc.
85  7. For PLACE actions and RELEASE actions, the robot must hold an object
       in the corresponding hand.
86  8. Before slicing an object, the robot can only interact with the object
       (e.g. peach_0), after slicing the object, the robot can only
       interact with the sliced object (e.g. peach_0_part_0).
87  9. You can only clean a stain with a soaked cleaning tool like rag, or
       put the stained object into a toggled on cleaner like sink or
       dishwasher.
88  10. To soak an object, first place the object into a toggled on sink,
       then soak it. Do not soak an object outside a sink.
89  11. Jars, Bags, and other objects must be OPENED, before you put things
       inside them.
90
91
92  Please output a SINGLE action command(in the given format) that the
       robot may execute next in order to make progress towards achieving
       the target environment state.
```

Now, the prompt given at each timestep

```
1   Your Task:
2   Input:
3
4   Currently, the robot is holding:
5   {robot_state}
6
7   Current environment state:
8   {object_state}
9
10  This may be summarized as the following atomic propositions being true:
11  {', '.join(APs) if APs else 'No atomic propositions are true.'}
12
13  Goal State description:
14  {task_description}
15
16  Feedback on failed actions from the environment:
17  {feedback if feedback else "No feedback yet."}
18
19
20  Feedback from the critic:
21  {failed_actions}
22
23  The feedback includes instructions from your critic (via an LTL law with
       an explanation), which will block certain actions that they think
       will lead to failure. It also includes any failed actions you have
       tried to execute in the past.
24  If you fail an action, please use the feedback to guide your next action
       choice.
25  DO NOT REPEAT AN ACTION IF THE CRITIC HAS BLOCKED IT OR IF IT HAS FAILED
       BEFORE.
```

```
Inner Monologue:
{inner_monologue}

Previous Successful action:
{old_action}

Please output the A SINGLE ACTION COMMAND (in the given format), the
    current environment state will make progress towards the target
    environment state.
Only output the action command with nothing else.

Output:
```

**Critic prompts:** First, context prompt for trajectories:

```
You are an expert symbolic critic analyzing a robot's task execution
    trajectory. '
Your goal is to propose Linear Temporal Logic (LTL) laws that will
    improve the robot's
efficiency, prevent common mistakes, and ensure task completion.

## ROBOT'S GOAL
{rule_data}

This goal is written in terms of a list of formulas involving APs. All
    of these formulas need to be true in order to finish the task.

## ATOMIC PROPOSITIONS

### Observation Variables (Environment State):
{chr(10).join([f"  - {ap}" for ap in APs['obs_APs']])}
**Key Observation Categories:**
- Object locations: `object_X_in_location`, `object_X_on_Y`,
    `object_X_inside_Y`
- Hand states: `object_X_in_hand` (what robot is holding)
- Object properties: `object_X_is_open`, `object_X_is_clean`, etc.
- Spatial relations: `object_X_next_to_Y`, `object_X_under_Y`

### Action Variables (Robot Actions):
{chr(10).join([f"  - {ap}" for ap in APs['action_APs_true']])}

**Action Categories:**
1. **Direct object actions**: `action_object` (grasp, open, close,
    clean, freeze, unfreeze, slice, soak, dry, toggle_on, toggle_off)
2. **Placement actions**: `object1_place_relation_object2` (place_ontop,
    place_inside, place_nextto, place_under, release)
3. **Transfer actions**: `object1_transfer_contents_relation_object2`
4. **Complex placement**: `object1_place_nextto_ontop_object2_object3`
5. **Task completion**: `done`

## LTL SYNTAX RULES
- **Operators**: `&` (and), `|` (or), `!` (not), `G` (globally), `X`
    (next), `->` (implies)
- **Format**: `G(observation_condition -> X(action_condition))`
- **Focus**: Prefer blocking bad actions rather than forcing specific
    actions
- **Trace structure**: (obs, action, obs, action, ...)

## YOUR TASK
Analyze the robot's trajectory and propose LTL laws that:
```

```
1. **Prevent inefficiencies**: Stop redundant or counterproductive
     actions
2. **Ensure prerequisites**: Block actions when preconditions aren't
     met. ( for example, a fridge must be open to place something inside
     it or take something out of it)
3. **Promote task completion**: Add rules to recognize when goals are
     achieved
4. **Maintain feasibility**: Avoid over-constraining the action space
```

Next, the critic main prompt for trajectories:

```
## TRAJECTORY ANALYSIS

You have already designed a few rules, however they were not enough to
     accomplish the task. You need to add an additional number of rules
     to get there!

### Robot Goal:
{rule_data}

### Execution Trace for the UNSUCCESSFUL RUN:
{format_AP_log_for_critic(AP_log)}

### Previous Rules:

Previously, you have already designed some rules based on priot traces,
     now, given a new trace, suggest a small number of ADDITIONAL RULES

The previous rules are:
{existing_rules}

## ANALYSIS FRAMEWORK

**Step 1: Identify the most repeated action in the trajectory
**Step 2: Understand why this action was repeated, and why it is
     necessary to repeat this action
**Step 3: Define LTL Laws which block this action when it is unnecessary

## EXAMPLES OF GOOD LTL LAWS

**Prerequisite checking:**
```
G(object_X_in_hand -> object_X_place_in_target_position)
"Only place objects you're actually holding"
```

**Efficiency enforcement:**
```
G((task_complete_for_X) -> X(!grasp_object_X))
"Don't grasp objects that are already correctly placed"
```

The goal consists of many parts as there are many objects in the
     environment.
You are refining an existing trajectory, focus on eliminating repeated,
     useless actions.
Do not constrain yourself to a small number of laws, make as many laws
     as you need. These laws are boolean, so be very precise.
Make different laws about different items. Dont try to merge all your
     laws into one big law, write many SIMPLE laws

```

```
## OUTPUT REQUIREMENTS

Provide your analysis in exactly this format:

**Explanation:**
1. **Initial State**: Describe the starting configuration across all
    relevant objects
2. **Goal Interpretation**: What the robot needs to accomplish for all
    listed sub-goals (treat them as possibly dependent)
3. **Required Steps**: Logical sequence to achieve the full multi-object
    goal
4. **Most repetitive action** : What was the most repetitive action in
    the trajctory provided?
5. **Law Strategy**: Propose a law to block this action when not
    necessary

**Laws:**
```json
[
{{
    "rule": "G(observation_condition -> X(action_condition))",
    "explanation": "Clear explanation of why this law improves
        performance"
}},
{{
    "rule": "G(another_condition -> X(another_action))",
    "explanation": "Another law addressing a different issue"
}}
]
```

**CRITICAL REMINDERS:**
- Use ONLY the provided observation and action APs
- Laws should be in format: `G(obs_condition -> X(action_condition))`
- Focus on blocking problematic actions, not forcing specific ones
- Ensure laws don't make the task impossible by over-constraining
```

Next, the critic context prompt for overconstrained states:

```
You are an expert symbolic critic observing a robot's behavior.

The robot's goal is:
{AP_log[-1]['goal']}

You are given:
1. A list of all atomic observation and action variables
2. The observation variables true at the current timestep
3. A set of LTL rules that are overconstraining (they block all actions)
4. The actions currently allowed by those laws (conflicting actions)

Your task:
- Analyze why the constraining rules conflict.
- Replace ONLY the constraining rules with new ones that resolve the
    deadlock.
- Keep all other rules unchanged.
- New rules must enforce **sequentiality** by adding conditions like
    `!o2` to break ties.
- New rules must strictly follow this format:
`G(expression1 -> X(expression2))`

Allowed operators:
- & (and), | (or), ! (not), G (globally), X (next), -> (implies)

Important:
- Each output must be valid JSON.
```

```
25   - Each rule must have the structure: {{"rule": "...", "explanation":
        "..."}}
26   - Output must be a JSON array of objects, with **double quotes only**.
27   - Do not output anything except the JSON array.
28
29   Example of correction:
30   If both rules are `G(o1 -> X(a1))` and `G(o2 -> X(a2))` and both o1, o2
        hold,
31   replace one with `G(o1 & !o2 -> X(a1))` and keep the second as G(o2 ->
        X(a2)).
32   Make sure to return both.
33
34   Think carefully about the goal and current state first.
35   Then output the replacement rules as JSON only.
```

Next, the critic main prompt for overconstrained states:

```
1    Observation variables:
2    {",".join(APs['obs_APs'])}
3
4    Action variables:
5    {",".join(APs['action_APs'])}
6
7
8    True observation variables at current timestep:
9    {AP_list_curr}
10
11   Overconstraining rules (to be replaced):
12   {constraining_rules}
13
14   Conflicting actions:
15   {valid_actions}
16
17   Now output the replacement rules in the following strict JSON format:
18
19   [
20   {{
21       "rule": "G(... -> X(...))",
22       "explanation": "..."
23   }},
24   {{
25       "rule": "G(... -> X(...))",
26       "explanation": "..."
27   }}
28   ]
29
30   Nothing else.
```

### A.6.2 MINECRAFT

**Actor prompts**  First, the context prompt:

```
1    You are a helpful assistant that responds with a primitive (built in
        mineflayer) which will lead to completing any Minecraft task
        specified by me.
2
3    At each round of conversation, I will give you
4    Code from the last round: ...
5    Execution error: ...
6    Chat log: ...
7    Biome: ...
8    Time: ...
9    Nearby blocks: ... ( A list of all uniqque blocks in a 16 block radius,
        you may use mineBlock to collect any of these blocks)
```

```
Nearby entities (nearest to farthest):
Neighbourhood blocks: ... (A list of blocks in your immediate
    neighbourhood i.e. a 2 block radius)
Health: ...
Hunger: ...
Position: ...
Equipment: ...
Inventory (xx/36): ... (A list of all items in your inventory, with
    their counts)
Chests: ...
Task: ...
Context: ...
Critique: ...
Previous failed code: ...

You should then respond to me with

Thinking:
Think out loud in natural language about what you observe, what you need
    to accomplish, and what you should do next. This should be free-form
    reasoning, not a structured list.

Code:
1) You must respond with a single line of code that corresponds to one
    of the following primitives:
    - Use `mineBlock(bot, name)` to collect blocks. Do not use `bot.dig`
        directly.
    - Use `craftItem(bot, name)` to craft items. Do not use `bot.craft`
        or `bot.recipesFor` directly.
    - Use `smeltItem(bot, itemName, fuelName)` to smelt itemName using
        fuelName. Do not use `bot.openFurnace` directly. Each item will
        consume one fuel.
    - Use `placeItem(bot, name, position)` to place blocks. Do not use
        `bot.placeBlock` directly.
    - Use exploreUntil(bot, direction, maxTime, callback) to explore,
        where,
        - direction is a Vec3 with values -1, 0, or 1 (e.g., new Vec3(1,
            0, 1) to explore diagonally).
        - maxTime is in seconds (default is 60).
        - callback is a function that returns a truthy value when the
            exploration goal is met. If it returns something truthy, the
            bot stops exploring early and exploreUntil returns that
            value. Otherwise, exploration continues until the time runs
            out. For example, callback can be () => {{
                    return bot.findBlock({{ matching: block =>
                        block.name === "iron_ore", maxDistance: 32 }});
                }}
    - Use `equipItem(bot, name, destination)` to equip an item in the
        bot's hand or armor slots. The default for destination is
        'hand'. For example, `equipItem(bot, "wooden_pickaxe")` equips a
        wooden pickaxe in the bot's hand.
2)  Every primitive function must be awaited, as they are asynchronous.
3)  Functions in the "last chosen primitive" section will not be saved
    or executed. Do not reuse functions listed there. If there is no
    error, it was executed successfully, if there is an error, it was
    not executed successfully
4) `maxDistance` should always be 32 for `bot.findBlocks` and
    `bot.findBlock`. Do not cheat.
5)  Do not use `bot.on` or `bot.once` to register event listeners. You
    definitely do not need them.
6)  Make sure you use the correct names for blocks and items, as they
    are case-sensitive. For example, use "stone" instead of "Stone",
    "oak_log" instead of "Oak Log", etc.
```

```
48
49  You should only respond in the format as described below:
50  RESPONSE FORMAT:
51  Thinking:
52  [Free-form reasoning about what you observe, what you need to do, and
        what action to take next]
53
54  Code:
55  ```javascript
56  await yourChosenPrimitive(bot,corresponding arguments);
57  ```
```

Now, the prompt given at each timestep

```
1   Response from the last round: \n {state.responseLastRound} \n
2   Execution error: {state.executionError if state.executionError else
        "None"}
3   Biome: {state.biome}
4   Time: {state.time}
5   Nearby blocks: {state.nearbyBlocks if state.nearbyBlocks else "None"}
6   Nearby entities (nearest to farthest): {", ".join([f"{entity.name}
        ({entity.type})" for entity in state.nearbyEntities]) if
        state.nearbyEntities else "None"}
7   Neighbourhood blocks: {", ".join([f"{block.name} at ({block.position.x},
        {block.position.y}, {block.position.z})" for block in
        state.neighbourhood]) if state.neighbourhood else "None"}
8   Health: {state.health}
9   Hunger: {state.hunger}
10  Position: ({state.position.x}, {state.position.y}, {state.position.z})
11  Equipment: Hand: {state.equipment.hand}, Armor: [Head:
        {state.equipment.armor.head}, Chest: {state.equipment.armor.chest},
        Legs: {state.equipment.armor.legs}, Feet:
        {state.equipment.armor.feet}]
12  Inventory (count: {state.inventoryCount}): {", ".join([f"{item.name}
        ({item.count})" for item in state.inventory]) if state.inventory
        else "None"}
13  Chests: {", ".join(state.chests) if state.chests else "None"}
14  Task: {state.task}
15  Context: {state.context}
16  Critique: {state.critic}
17  Previous failed code: You previously attempted the following codes, and
        they didnt work because they violated the critics recommendation
        {failed_codes if failed_codes else "None"}
```

Next, the critic context prompt for trajectories:

```
1   You are an expert critic observing the trajectory of a Minecraft agent.
        The goal of the agent is to mine a diamond.
2
3   You are given:
4   1) A list of atomic observation and action variables
5   2) A list of failures that occurred in trajectories
6   3) Existing LTL rules implemented
7
8   You will be given a series of steps taken by the agent, including
        observations, actions, and success/failure of the action with an
        error message.
9   Your task is to analyze the trajectory and provide LTL laws that
        constrain the agent's actions in order to boost efficiency and
        performance.
10  The laws should be in the form of LTL formulas, and you should provide a
        brief explanation of each law. The boolean variables used in the
        laws are defined as follows:
11
```

```
12  Observation Variables:
13  {", ".join(OBS_VARIABLES_LIST)}
14
15  The observations in the atomic proposition space are described as
        follows:
16
17  - `obs_has_x` corresponds to having the item `x` in the inventory of the
        agent.
18  - `obs_near_crafting_table` or `obs_near_furnace` define if the agent is
        within an interacting distance of a crafting table or furnace.
19  - `obs_has_x_equipped` corresponds to an object `x` (e.g., an iron
        pickaxe) actively equipped.
20  - Only one item can be equipped at any point in time.
21  - You may propose additional observation variables if needed to express
        useful rules.
22
23  Action Variables:
24  {", ".join(ACTION_VARIABLES_LIST)}
25
26  The actions the agent can perform are limited to a few types:
27
28  - `action_mine_x`: mines the item `x`. Certain blocks require certain
        tools:
29      - stone: wood pickaxe or better
30      - iron: stone pickaxe or better
31      - diamond: iron pickaxe or better
32  - `action_craft_x`: crafts an item `x`, if prerequisites and (if needed)
        a crafting table are present.
33  - `action_smelt_iron`: smelts raw iron into ingots using fuel and a
        furnace.
34  - `action_equip_x`: equips a tool for mining. Only one tool can be
        equipped at a time.
35  - `action_explore`: used to find resources not currently visible.
36  - `action_place_x`: places an item like a crafting table or furnace to
        enable usage.
```

Next, the critic main prompt for trajectories:

```
1   You are a symbolic critic observing the trajectory of a Minecraft agent.
        The agent is inefficient, often repeats work, and occasionally
        causes errors like trying to mine without the right tool or crafting
        without the ingredients.
2
3   GOAL OF AGENT: MINE A DIAMOND
4   YOUR GOAL: PROPOSE LTL LAWS THAT PROMOTE THE AGENT'S PROGRESS TOWARD
        THIS GOAL AND PREVENT INEFFICIENCIES.
5
6   THINK OF THE BASIC TASK GRAPH REQUIRED TO MINE A DIAMOND, and PROPOSE
        LTL LAWS TO GUIDE THE AGENT ALONG THAT GRAPH.
7
8   Before forcing any action, think of checking if the subgoals to do that
        action are met.
9
10
11  ### Your task:
12  1. Decompose the task of mining a diamond into symbolic subgoals:
        acquiring wood, crafting tools, smelting, equipping tools, etc.
13  2. For each subgoal transition (e.g., "has stone => craft
        stone_pickaxe"), propose an **LTL law** that enables or encourages
        this step.
14  3. Also identify any **errors** or **inefficiencies** in the trajectory.
        For each one, propose an LTL law to prevent that mistake in the
        future.
```

```
4. Focus on writing LTL laws in the form `G(condition => X(action))`.
    Use observation variables for `condition`, and action variables for
    `action`.
5. Avoid overly specific or redundant laws. Try to generalize from the
    plan, not just from individual steps.
6. You may also propose **new boolean observation variables** if needed
    to express useful constraints (e.g., `obs_has_stone_pickaxe`,
    `obs_seen_diamond_block`).
7. Your final set of LTL laws should include:
- >=3 rules that **encourage efficient, goal-aligned behavior**
- >=1 rule that **discourages observed inefficient behavior**

### Existing Inputs:
- Existing LTL laws: {SOFT_LTL_RULES_SAYCAN}
- Action and observation variables:
    - Actions: {", ".join(ACTION_VARIABLES_LIST)}
    - Observations: {", ".join(OBS_VARIABLES_LIST)}

### Agent Trajectory:
{format_trajectory_for_critic(trace)}

### Output Format:
Return the following in your response:

---

### Reasoning:
1. **Plan Decomposition**: Write out the full high-level plan to mine a
    diamond as a sequence of symbolic subgoals (e.g., get wood-> make
    planks-> craft tools-> smelt-> equip-> mine).
2. **Plan  Conversion**:  List the sequence of obs props and action
    props that correspond to this plan
2. **Positive Constraints**: For at least four transitions in this plan,
    propose a rule of the form `G(preconditions => X(useful_action))`
    that helps the agent complete the task efficiently.
3. **Negative Constraints**: Identify any mistakes in the trajectory
    (e.g., crafting without ingredients, mining without tools), and
    write `G(bad_condition => X(!bad_action))` rules to prevent them.
4. **Coverage**: Ensure your rules cover multiple stages of the plan
    (not just early or late stages).
5. **Reusability**: The rules should generalize and not rely on specific
    step numbers.

---
Laws:
- `efficiency_laws`: a list of LTL rules that **promote efficient
    behaviors**, each with a brief explanation.
- `inefficiency_laws`: a list of LTL rules that **prevent mistakes**,
    each with a brief explanation.
- Mention which part of the plan each law corresponds to.
```

Next, the critic context prompt for overconstrained states:

```
You are an expert critic observing the trajectory of a Minecraft agent.
    The goal of the agent is to mine a diamond.

Sometimes, the laws you impose are too constraining and prevent all
    possible actions. Your job is to break these deadlocks

You are given:
1) A list of atomic observation and action variables

```

```
8   You will be given a particular timestep where the LTL laws led to no
        feasible action, and your task is to resolve the conflict by either
        modifying or deleting one of the LTL laws.
9
10  The laws should be in the form of LTL formulas, and you should provide a
        brief explanation of each law. The boolean variables used in the
        laws are defined as follows:
11
12  Observation Variables:
13  {", ".join(OBS_VARIABLES_LIST)}
14
15  The observations in the atomic proposition space are described as
        follows:
16
17  - `obs_has_x` corresponds to having the item `x` in the inventory of the
        agent.
18  - `obs_near_crafting_table` or `obs_near_furnace` define if the agent is
        within an interacting distance of a crafting table or furnace.
19  - `obs_has_x_equipped` corresponds to an object `x` (e.g., an iron
        pickaxe) actively equipped.
20  - Only one item can be equipped at any point in time.
21  - You may propose additional observation variables if needed to express
        useful rules.
22
23  Action Variables:
24  {", ".join(ACTION_VARIABLES_LIST)}
25
26  The actions the agent can perform are limited to a few types:
27
28  - `action_mine_x`: mines the item `x`. Certain blocks require certain
        tools:
29      - stone: wood pickaxe or better
30      - iron: stone pickaxe or better
31      - diamond: iron pickaxe or better
32  - `action_craft_x`: crafts an item `x`, if prerequisites and (if needed)
        a crafting table are present.
33  - `action_smelt_iron`: smelts raw iron into ingots using fuel and a
        furnace.
34  - `action_equip_x`: equips a tool for mining. Only one tool can be
        equipped at a time.
35  - `action_explore`: used to find resources not currently visible.
36  - `action_place_x`: places an item like a crafting table or furnace to
        enable usage.
```

Next, the critic main prompt for overconstrained states:

```
1   You are a symbolic critic observing the trajectory of a Minecraft agent.
        The agent is inefficient, often repeats work, and occasionally
        causes errors like trying to mine without the right tool or crafting
        without the ingredients.
2
3   GOAL OF AGENT: MINE A DIAMOND
4   YOUR GOAL: You are given a timestep where the LTL laws led to no
        feasible action, and your task is to resolve the conflict by either
        modifying or deleting one of the LTL laws.
5
6   Given the set of observations, no actions are allowed in the given
        instance. Modify one or both of the laws to break this deadlock.
7
8   ### Your task:
9   1. First, reason about which rule is less useful given the constraints
10  2. Modify that law
11  3. Make sure there is atleast one feasible action in the given state
        after the modification of laws.
12
```

```
13  ### Rules you should output:
14  - Each rule should follow the form: `G(condition => X(action))`
15  - Use **observation variables** (e.g., obs_has_x, obs_near_x,
        obs_equipped_x) in the `condition`.
16  - Use **action variables** (e.g., action_mine_x, action_craft_x) in the
        `action`.
17  - Conditions should reflect **states that actually occurred** in the
        trajectory, so the rule can generalize and not be overly specific or
        invalid. The corresponding action should either approve or
        disapprove of the corresponding action in the trajectory.
18  - Make sure the rules do not block **all** possible actions. The agent
        always needs at least one valid option.
19  - Make sure to state which timesteps your rule is based on.
20  - If necessary, propose **new observation variables** that could help
        express useful rules.
21
22
23
24  ### Format:
25  - Output the LTL laws as a list of strings, each in proper syntax
26  - Then provide a brief explanation of each rule and how it prevents
        error or improves efficiency
27  - Output as many laws as you deem necessary, forcing efficient actions
        and disallowing inefficient ones
28
29  ### Example LTL Law:
30  G(obs_has_raw_iron ^ obs_near_furnace => X(action_smelt_iron))
31  Explanation: Smelting iron early helps the agent craft a better pickaxe
        sooner.
32
33  ### Inputs:
34
35  - Existing LTL laws
36      Rule 2: G(obs_has_2x_plank & !obs_has_2x_stick ->
            X(action_craft_stick))
37      Number of actions allowed: 1
38      Filtered actions:
39      ['action_craft_stick']
40      Observation Propositions (obs_props):
41      obs_has_plank: True
42      obs_has_2x_plank: True
43      obs_has_3x_plank: True
44      obs_has_wood_pickaxe: True
45      obs_has_stone_pickaxe: True
46      obs_has_fuel: True
47      obs_near_crafting_table: True
48      obs_iron_in_chunk: True
49      obs_coal_in_chunk: True
50      obs_wood_pickaxe_equipped: True
51      ==================================================
52      Rule 7: G(obs_wood_pickaxe_equipped & !obs_has_3x_cobble ->
            X(action_mine_stone))
53      Number of actions allowed: 1
54      Filtered actions:
55      ['action_mine_stone']
56      Observation Propositions (obs_props):
57      obs_has_plank: True
58      obs_has_2x_plank: True
59      obs_has_3x_plank: True
60      obs_has_wood_pickaxe: True
61      obs_has_stone_pickaxe: True
62      obs_has_fuel: True
63      obs_near_crafting_table: True
64      obs_iron_in_chunk: True
65      obs_coal_in_chunk: True
```

```
66      obs_wood_pickaxe_equipped: True
67
68 These rules are too constraining, modify them.
69
70
71 - Feasible actions per step are available (so do not block everything)
72 - Action and observation variables:
73     - Actions: {", ".join(ACTION_VARIABLES_LIST)}
74     - Observations: {", ".join(OBS_VARIABLES_LIST)}
75
76
77 Answer with the following three things
78
79 1. Identify the action the agent should take given this state
80 2. Identify which rules are blocking that action from happening
81 3. Modify those rules.
82
83
84 Modify one of the rules, or delete one of them, so that the agent can
        take a feasible action at this timestep.
```

## A.7 CODE AND TRAJECTORIES

The code, trajectories, and the LTL laws generated for Behavior are in the supplementary material.