# OpenReview forum: "LogicGuard: Improving embodied LLM agents through temporal logic based critics"
_ICLR.cc/2026/Conference — Submitted to ICLR 2026_

### Official Review · Reviewer_naM6 · 2025-10-20

**Soundness:** 3
**Presentation:** 3
**Contribution:** 2
**Rating:** 6
**Confidence:** 4

**Summary:**

The authors propose an actor-critic method consisting of two parts. An LLM-actor, which can be any LLM-based planner, and an LLM-critic, called LogicGuard. LogicGuard uses trajectory history to infer rules in a reactive fragment of LTL, which are then used in conjunction with a verifier to allow the actor’s actions to proceed or to provide feedback to the actor. The proposed method is evaluated in two domains with two different planners, showing an increase in task completion and efficiency over implementations that do not use LogicGuard.

**Strengths:**

This work tackles an important question and is mostly well-written. This is a very crowded area and the authors have taken great pains to make clear the distinction between this paper and related work.

The proposed architecture makes a lot of sense, and the fact that it contains a verifier after *both* LLMs helps avoid the problem of “LLMs all the way down.”

It is nice to see efficiency gains as well as safety improvements in the results.

**Weaknesses:**

The authors overstate the application of LTL in this work. The work uses a *fragment* of LTL, that is purely reactive. That is, all specifications are of the form “Always preconditions implies next postconditions”. This may be a useful fragment, and indeed the results show some positive effect. But the authors should be very clear up front about the fact that their work is for a very small fragment of LTL.

The graph planning portion is not very clear to me. It is quite common in LTL to use graph traversal for planning, even with reactive fragments. See for example [1] for incremental graph construction and pruning, and [2,3] for reactive synthesis as a graph game. How does the graph approach compare to these? Is there any novelty to this part compared to standard work in LTL?

I’m not sure the communication via LTL is entirely novel. There is plenty of work in automatically generating LTL specifications from examples using LLMs. For example, [4] and [5]. The authors should comment on the novelty of this aspect.

[1] Vasile CI, Li X, Belta C. Reactive sampling-based path planning with temporal logic specifications. The International Journal of Robotics Research. 2020;39(8):1002-1028. doi:10.1177/0278364920918919

[2] Ehlers, R., Khalimov, A. (2024). “Fully Generalized Reactivity(1) Synthesis”. Tools and Algorithms for the Construction and Analysis of Systems. TACAS 2024.

[3] Bloem, Roderick, Krishnendu Chatterjee, and Barbara Jobstmann. "Graph games and reactive synthesis." Handbook of model checking. Cham: Springer International Publishing, 2018. 921-962.

[4] Vazquez-Chanlatte, Marcell, et al. "L LM: Learning Automata from Demonstrations, Examples, and Natural Language." (2025).

[5] Gupta et al. “Integrating Explanations in Learning LTL Specifications from Demonstrations”

**Questions:**

How is S_unsafe determined? Is it entirely inferred by the critic?

See above for questions about graph planning.

How necessary is the LLM as critic? It seems like something like [6] could be used to mine these specifications. Is there a reason it must be an LLM?

Can this method extend to more general forms of LTL specifications?

[6] Hasanbeig, Mohammadhosein, et al. "Deepsynth: Automata synthesis for automatic task segmentation in deep reinforcement learning." Proceedings of the AAAI conference on artificial intelligence. Vol. 35. No. 9. 2021.

---

> ### Author Response · Authors · 2025-11-17
> **Response to Reviewer naM6**
>
> We sincerely thank the reviewer for the thoughtful and detailed feedback, and for pointing us to several highly relevant references. We deeply appreciate the constructive tone and the educational nature of your comments. Please find below a response to each of your comments.
>
> **W1: Reactive LTL**
>
> We agree with the reviewer that our specifications belong to a reactive fragment of LTL, of the form, $G(\text{precondition} \rightarrow \text{postconditions})$, and we now clarify this explicitly in the revised manuscript in the subsection titled LTL primer. We focus on this fragment to enable real time verification during agent execution, as our agents only predict actions one step ahead. For this setup, this gives us sufficient expressivity to improve performance.
>
> **W2/Q2: Graph planning**
>
> We thank the reviewer for this helpful comparison, and pointing us towards this literature. To the best of our understanding, works such as [1–3] employ graph traversal and synthesis over known graphs with pre-specified automata. Our setting differs in two ways. (i) The environmental graph is large, unknown and must be discovered and traversed online through interaction and (ii) we wish to dynamically identify and prune infeasible actions while suggesting corrective ones, by preventing or forcing transitions through certain edges on the graph, rather than computing plans over a fixed, known graph structure. The main novelty of our work lies in this coupling, using the LLM’s semantic understanding to heuristically guide exploration in partially observed environments where the underlying graph is unknown and may be large or dynamically changing. This enables sample-efficient learning without requiring full graph construction or exhaustive search.
>
> **W3: Learning LTL from demonstrations**
>
> We thank the reviewer for highlighting [4, 5]. These works generate LTL specifications from demonstrations or multimodal inputs, typically in a one-shot manner to capture demonstrated behaviors. Our approach differs both in goal and mechanism: rather than inferring specifications from demonstrations, our LLM critic  iteratively analyzes failed rollouts to generate corrective constraints that prevent specific failure modes in future rollouts. This shifts the goal from specification inference to self-supervised symbolic refinement. We revise our related work section, and conclusion accordingly to more clearly distinguish LogicGuard from [4, 5].
>
> **Q1: Defining $\mathcal{S}_\text{unsafe}$**
>
> **$\mathcal{S}_\text{unsafe}$** is predefined by the user and encodes safety-critical conditions (e.g., collisions or invalid actions). The LLM critic  does not modify or infer these states; it only proposes performance-related constraints. In Minecraft using the SayCan baseline, unsafe states correspond to those reached after taking invalid or infeasible actions, while in our other settings this set is empty. This separation ensures that safety remains verifiable and independent of LLM behavior.
>
> **Q3: Replacing the LLM as a critic**
>
> We appreciate this suggestion. Methods such as [6] could in principle mine automata from demonstrations, but they typically require explicit exploration over a known or small state graph. In contrast, our environments involve large, partially observed, dynamically changing graphs, where such exhaustive exploration is intractable. The LLM critic provides a practical heuristic for identifying symbolic regularities from limited trajectories, making it suitable for long-horizon embodied tasks.
>
> **Q4: More general LTL specifications**
>
> We thank the reviewer for this question. Our framework focuses on reactive constraints, which can be verified using only the current and previous states, without requiring predictions of future states. This design choice offers important practical advantages: (1) constraints can be verified in real-time during execution without maintaining a world model, (2) we avoid the hallucination risks inherent in asking LLMs to predict future environment states, and (3) reactive constraints are sufficient to express many critical safety properties (e.g., avoid-dangerous-states, respect-resource-constraints).
>
> Extending to full LTL would require predicting how actions affect future observations. This would necessitate either a learned world model (introducing model error and hallucination risks) or explicit environment simulators (limiting generality). While an important direction for future work, our use of the reactive fragment provides a balance between expressivity and reliability for online deployment.

---

### Official Review · Reviewer_uyWu · 2025-10-28

**Soundness:** 3
**Presentation:** 3
**Contribution:** 2
**Rating:** 4
**Confidence:** 3

**Summary:**

This paper proposes LogicGuard, a modular actor–critic framework where the actor is guided by a trajectory-level LLM critic, communicating through Linear Temporal Logic (LTL). The proposed method improves task completion rates on both short-horizon and long-horizon tasks.

**Strengths:**

1. The system combines symbolic reasoning with the generalization ability of LLMs.
2. Every constraint has a verbalized explanation, helping users inspect the agent’s decision process.

**Weaknesses:**

1. Offline critic analysis and frequent rule updates require repeated LLM calls, which can be expensive and potentially unstable.
2. This work hand crafts a large number of rules and complex prompts. This constraints the environment and lowers the task difficulty.

**Questions:**

1. Can learned LTL rules transfer across tasks or environments, or must they be relearned each time?

---

> ### Author Response · Authors · 2025-11-18
> **Response to Reviewer uyWu**
>
> We sincerely thank the reviewer for their thoughtful and constructive feedback, and for highlighting two of the core strengths of our work: LogicGuard’s interpretability and its hybrid design that merges the complementary benefits of symbolic reasoning and LLM generalization. We truly appreciate your engagement with the paper, and your comments have helped us identify places where our exposition can be strengthened.
>
> **W1: Offline critics**
>
> Thank you for raising this important point. Our design choice to run the critic offline is precisely motivated by the concerns you mention. Running the critic online would indeed trigger repeated LLM calls and unstable rule oscillations. Instead, we observe that the offline critic converges extremely quickly. Typically within 1–2 iterations on Behavior and 2–5 on Minecraft, depending on the base policy. This small, bounded number of updates ensures both stability and cost-efficiency, even on long-horizon tasks.
>
> Following another reviewer’s suggestion, we also implemented an LLM-as-a-judge baseline biased by the same rules discovered by our critic, but free to operate as an online critic. Consistent with your intuition, this online alternative proved far more expensive, less stable, and prone to severe inconsistencies, further validating the practicality of our offline approach.
>
>
> **W2: Handcrafting of rules and prompts**
>
> We respectfully clarify that the amount of handcrafting in LogicGuard is much smaller than it may appear. The only manually specified rules appear in the SayCan baseline, which inherently requires a predefined affordance function as introduced in [1]. Beyond this, LogicGuard’s rule synthesis is _fully automatic_.
> * In Behavior, we use a single, task-agnostic, environment-agnostic prompt across 100 diverse tasks and environments.
> * In Minecraft, our prompt is domain-tailored but comparable in complexity to prior LLM-Minecraft agents such as Voyager[2].
>
> Any complexity in our prompt design serves to bias the LLM toward generating rules that can be efficiently validated by our programmatic verifier. This reduces redundant LLM queries and does not simplify the underlying task.”
>
> [1] Ahn et al, "Do as i can, not as i say: Grounding language in robotic affordances", Arxiv
>
> [2] Wang et al, "VOYAGER: An Open-Ended Embodied Agent with Large Language Models", TMLR 2024.
>
>
> **Q1: Transfer of LTL rules**
>
> We appreciate this insightful question. At present, rules are environment and task specific because of the diversity in the underlying tasks being performed, environments, and objects in the environment. However, we may readily extend the critic LLM to support subtask-level tagging, enabling transfer of rules whenever the same skill or subtask appears across environments.
>
> This establishes a clear path toward compositional rule transfer, where reusable subtasks naturally bring their associated constraints with them. This is also an interesting avenue for future work if there is an environment where subtasks repeat frequently.
>
>
>
> We thank the reviewer once again for their thoughtful and constructive feedback. We hope these clarifications highlight that LogicGuard is both stable and minimally hand-engineered, while offering a practical, interpretable framework that scales across diverse tasks. Your comments have helped us improve the clarity of our presentation

---

### Official Review · Reviewer_67BB · 2025-10-28

**Soundness:** 3
**Presentation:** 4
**Contribution:** 2
**Rating:** 6
**Confidence:** 5

**Summary:**

LogicGuard proposes a modular actor-critic architecture where an LLM actor generates high-level actions while a trajectory-level LLM critic communicates constraints via Linear Temporal Logic (LTL). The system addresses compounding errors in long-horizon sequential planning by automatically generating formal constraints that guide LLM-based agents. Evaluated on Behavior-100 (household tasks) and Minecraft diamond-mining, LogicGuard achieves 25% improvement in task completion rates and 23% efficiency gains over baseline InnerMonologue, while enabling SayCan to complete previously impossible tasks. This work presents itself as actor-critic framework but this work appears as LLM as Jude. Paper is written very nicely and enjoyed reading the paper.

**Strengths:**

1. Novel Integration of Formal Methods with LLMs:
The use of LTL as a communication protocol between actor and critic is innovative and addresses a critical gap in LLM-based planning. Prior work using natural language feedback, LTL provides verifiable, machine-checkable constraints with formal guarantees but this work went ahead to have a online actor critic.

2. Strong Empirical Results Across Diverse Settings:
The paper demonstrates substantial improvements in both generalist (Behavior: 47%->72% completion) and specialist settings (Minecraft: 23% efficiency gain, 23%->4.5% failure rate). The fact that SayCan completely fails without LogicGuard (0/5 success) but succeeds consistently with it (5/5) is particularly compelling.

3. Modular and Model-Agnostic Design:
The architecture allows direct integration with existing LLM planners (InnerMonologue, SayCan) as a "logic-generating wrapper," enhancing practical applicability. The separation of online actor and offline critic loops is well-motivated.

4. Theoretical Grounding:
Theorem 1 bounds the number of possible LTL rules generated from trajectories, providing formal guarantees about constraint space complexity. The graph-theoretic formulation of planning as bipartite graph traversal is elegant

5. Automatic Atomic Proposition Generation:
The automated AP generation for Behavior dataset demonstrates scalability potential, though hand-engineered APs remain necessary for specialist tasks

**Weaknesses:**

1. Missing ablations and baselines:There is no direct comparison between online vs. offline critic modes, which is crucial to understand trade-offs in adaptability and computational overhead. Baseline coverage (e.g., other LLM-based critics or symbolic guardrail methods) is limited. What is the contribution of each critic source (environment feedback, graph-based efficiency, over-constrained states)?

2. Evaluation scale and statistical rigor: Reported experiments appear based on small sample sizes (e.g., 5 trials for Minecraft) and lack confidence intervals or significance testing. Reported gains, while large, are not statistically supported.

3. Sensitivity to AP design: The system heavily relies on the quality and coverage of atomic propositions (APs). The paper itself acknowledges failures in auto-generated APs but provides no quantitative analysis or robustness study.

4. Limited theoretical contribution: Theorem 1 offers only a trivial bound on rule count rather than a meaningful performance or safety guarantee. Either the analysis should be expanded or reframed more modestly.

(Minor):

1. Theorem 1 bounds constraints at O(N) where N is trajectory length, but exponential state space (2^n) remains problematic.

**Questions:**

**(General Note:)** 1.  The proposed framework appears to function more as an LLM-as-Judge, where the model evaluates and corrects its own reasoning without a learning loop or external reward signal. Could the authors clarify why this is framed as an actor-critic setup rather than a judgment-based architecture? The distinction affects how we interpret its generalizability and learning dynamics.

2. Can you report comparisons between the offline critic (used in this work) and an online critic that generates or enforces constraints during rollout? This would clarify whether the offline setup is a performance vs. efficiency trade-off.

3. How sensitive is LogicGuard’s performance to AP quality? For example, if APs are noisy, missing, or partially wrong, how often do induced rules misfire or block safe actions?

4. Could you clarify how over-constrained or conflicting laws are detected and pruned? The threshold-based deletion policy is mentioned but not analyzed quantitatively.

5. Please provide statistical metrics (mean ± CI) for Behavior and Minecraft results . This would make the empirical section much stronger.

6. Please comapre and clarify how LogicGaurd differs from work like Veriplan [1], CotTL[2]. As for example veriplan and other recent work on formal verification for LLM planning would help situate LogicGuard among existing logic-verification frameworks, although not on actor critic side. Similarly Cot-TL also employs automata-based temporal checking to verify plans, similar to your LTL critic.

7. The critic’s induced laws are interpretable, but how scalable is this approach when the number of APs grows large (e.g., combinatorial explosion)?

 8. Constraint Accumulation: Do LTL constraints accumulate over time? Is there constraint pruning? The adaptive removal mechanism (triggered beyond fixed threshold) is mentioned but not evaluated. IN case I missed it, please point out relevent section.

9. SayCan baseline uses hand-engineered affordance function while LogicGuard versions use critic-generated laws - inconsistent comparison. Please clarify or did I missed something.

[1] VeriPlan: Integrating Formal Verification and LLMs into End-User Planning

[2] CoT-TL: Low-Resource Temporal Knowledge Representation of Planning Instructions Using Chain-of-Thought Reasoning

---

> ### Author Response · Authors · 2025-11-18
> **Response to Reviewer 67BB Part 1: Baselines, Ablations and Critic modes**
>
> We are glad to hear that the reviewer enjoyed reading our paper, and that they appreciated our modular architecture, our novel interaction of LTL formalism with LLM based heuristics. We adress your comments in three parts.
>
> ## Ablations, Baselines, and Architectural Clarifications (W1, Q1, Q2)
>
> **Ablations and Online–Offline Comparisons.**
>
> We thank the reviewer for drawing the connection to LLM-as-a-Judge architectures. In the revision, we add ablations that interpolate between LogicGuard and a LLM-as-a-Judge, enabling direct comparison of (i) online vs. offline critics, and symbolic vs. LLM-based verification. We introduce two ablations.
>
> 1) **LLM Verifier**: We replace the LTL verifier with an LLM judge, who is given by the offline generated LTL laws, but can choose to block or allow any action at its will.
>
> 2) **Full LLM-as-a-Judge**: Originally, the actor's prompt is biased with a filtered list of legal actions per current LTL laws. We eliminate this part of the prompt, as well as replace the LTL verifier with an LLM, yielding a complete LLM as a judge setup.
>
> These variants place the LLM in an online critic role, biased by the same law set produced offline to ensure a fair comparison. They yield comparable but slightly worse mean performance, with higher variance, frequent misinterpretations (11–12%), occasional catastrophic failures, and significantly greater computational overhead. In contrast, LogicGuard’s offline critic + symbolic verifier is stable, predictable, and runs in real time. This aligns with our goal of trustworthy, verifiable planning. Details and results are included in a new ablation section in the results and the appendix.
>
> **Actor-Critic vs LLM-as-a-Judge**
>
> We added a short discussion in the main text clarifying this distinction. In summary, LLM-as-a-Judge systems rely on repeated online LLM queries with no formal checking, operating as stochastic black boxes that may misinterpret environment state or constraints. LogicGuard instead uses the LLM judge only once per training iteration to synthesize machine-readable laws that are verifiably grounded in trajectory data and enforced efficiently by a symbolic verifier. The critic is therefore offline and produces structured, executable rules, while the symbolic component plays the role of a runtime critic enforcing them. This modularity yields predictable and reproducible behavior and avoids the computational and safety bottlenecks inherent to online judgment-based systems.
>
> **Contribution of Critic Signals**
>
> Environment feedback and graph-based efficiency patterns are jointly generated through the critic prompt and cannot be decoupled reliably, without manually checking and subjective classification, since avoiding a failed action technically leads to more efficiency by saving an action. The only independently switchable component is the over-constrained–state filter, whose removal always yields deadlocks, since all actions are blocked. The relative influence of each signal depends on the domain. Intuitively,
>
> - Minecraft: The environment produces many failure cues, the reduction in environmental feedback errors (which we term safety violations) are correlated with the improvements due to environment feedback
>
> - Behavior. The simulation environment provides almost no feedback in the form of an error. Most learned laws come from graph based efficiency laws.
>
> ## Evaluation Scale and Statistics (W2, Q5)
>
> Thank you for pointing out this omission. We have now added full statistical reporting (mean ± std and 95\% confidence intervals) for all Minecraft experiments in the revised manuscript. We direct the reviewer to the updated paper for the complete results.
>
> **Minecraft (5 trials per setting).**
>
> Despite the small number of trials, the performance differences between LogicGuard and the baselines are large and consistent across all seeds, leading to statistical significance. We also include statistics for the ablation baselines, but we do not claim significance there, as the differences are smaller, and can be explicitly traced to a few systemic errors which we present in our appendix.
>
> **Behavior (100 tasks).**
> On all 100 tasks,
> - Base actor:  47 ±9.8\%,
> - Logicguard: 72±8.8\%,
>
> where CI's are derived by treating task outcomes as Bernoulli trials. The intervals do not overlap, providing strong statistical evidence of improved task success. While confidence intervals for accuracy-type metrics are not commonly reported in related work, we can include them in the paper if the reviewer prefers.
>
> We appreciate the reviewer’s emphasis on rigorous reporting; the additional statistics strengthen our empirical evaluation.

---

> ### Author Response · Authors · 2025-11-18
> **Response to Reviewer 67BB Part 2: Constraint accumulation, AP Design, and Scalability**
>
> ## Constraint accumulation and pruning (Q4, Q8)
>
> We only deal with over constrained states whenever they are encountered in rollouts, as it is infeasible to check all (exponentially many) states. At runtime, if the agent reaches a state with no feasible actions, we automatically relax the active laws to the safety-only subset, ensuring progress. Offline, the LLM critic is provided with the set of laws responsible for over-constraining the current state (the “most constraining laws”), and selects one or more to delete or modify. This mechanism is described in Section 3.5.
>
> **Pruning of Constraints**
>
> The threshold-based deletion rule serves a different purpose: it prunes incorrect performance laws. If a non-safety constraint is violated three times by the actor, it is removed. Deleting a law on the first violation would collapse the critic entirely, while larger thresholds allow harmful rules to persist. Empirically, the threshold of three strikes balances stability and correction. We will add quantitative results reporting its effect in the camera-ready version.
>
> **Accumulation of Constraints**
>
> Yes, LTL laws accumulate across critic iterations, and we stop iterating once performance stabilizes or begins to degrade—analogous to early stopping to avoid overfitting.
>
>
> ## Scalability and Sensitivity to Atomic Propositions (W3, Q3, Q7)
>
> **Scalability with Large AP Sets.**
>
> While LLMs are known to struggle with combinatorial reasoning, our setup avoids this pitfall. Each action in our environment produces only local, incremental changes, so the critic is prompted only with the APs whose values flipped at the current timestep. This isolates causal effects and keeps the reasoning problem small, independent of the total AP set. For extremely long horizons, context limitations may appear, but strategies such as temporal chunking can mitigate this. Overall, scalability is driven by locality, not the total AP count.
>
> **Sensitivity to AP Quality: Noise, Missing APs, and Misfires**
>
> Safety-related APs are hand-engineered and form a fixed baseline set; these are not automatically generated and therefore cannot be corrupted by the critic. The LLM only operates within this safe set. In practical systems, sensor noise may induce errors in AP values, but this is a general engineering challenge rather than a limitation unique to LogicGuard. Since our runtime verifier enforces safety laws symbolically, safety is never delegated to the LLM and is therefore robust to noise from any hallucinations or state grounding.
>
> **Dependence on AP Design and Coverage.**
>
> We agree that the expressiveness of the APs significantly influences performance. This dependence is fundamental to any method relying on symbolic abstraction: APs in LogicGuard play the same role as features in RL or symbolic encodings in classical planning. Richer abstractions yield more informative laws. Our automatic AP-generation procedure already provides gains across Behavior despite being task and environment agnostic. An exciting avenue for future work is exposing the code base or environment metadata to the LLM so it can propose additional APs dynamically, although this needs to be carefully implemented.
>
> Robustness analysis over the discrete AP space is combinatorially infeasible and would require arbitrary task- and AP-specific categorizations. Instead, we emphasize the conceptual takeaway supported by our experiments: more expressive APs consistently improve performance, and LogicGuard remains effective even with automatically generated APs.

---

> ### Author Response · Authors · 2025-11-18
> **Response to Reviewer 67BB Part 3: Related work, theoretical implications, and minor comments**
>
> **Comparison with Related Work (Q6)**
>
> Thank you for pointing us to VeriPlan and CoT-TL. While these works also use LTL, their goals and mechanisms differ from LogicGuard in two key ways:
>
> 1) **Source of constraints.**
>
> VeriPlan and CoT-TL convert user-provided instructions into fixed LTL formulas. LogicGuard’s critic instead generates constraints from trajectory failures, capturing efficiency issues (e.g., redundant moves or unnecessary tool switches) absent in the task description.
>
> 2) **Role of Verifier**
> VeriPlan and CoT-TL perform one-shot plan verification. LogicGuard uses LTL in a feedback loop: the critic proposes constraints, the verifier enforces them, and the agent updates future plans. This enables adaptive refinement rather than static checking.
>
>
> We agree these papers are relevant on the verification side and have cited them as related work, while clarifying that LogicGuard focuses on designing LTL laws that improve efficiency, which to our knowledge is not addressed in existing logic-verification frameworks.
>
> **Theoretical Contributions and future avenues (W4,W5)**
>
> We thank the reviewer for this comment. In response, we have renamed Theorem 1 to Remark 1, as it provides a simple counting bound rather than a full performance or safety guarantee. We have also added a second remark, discussing how one may quantify the effect of each LTL law very precisely on the baseline policy. The authors believe that depending on the application, the use of formal logic enables one to claim many precise mathematical statements about blackbox LLM behavior, with strong implications on partner adaptation in multi agent teams, safety violations in stochastic environments, and other domains. We note that such statements are not possible for standard LLM-as-a-Judge systems, which are completely blackbox.
>
> Regarding Q9, via remark 2, we precisely quantify the search space within which the critic can modify the baseline policy. While this prevents us from making any statements about global optimality of the final trajectory, we reduce a very large search space of policies to a tube around the current policy, hopefully leading to iterative improvements. In general, due to the exponential size of the state space, finding the optimal trajectory is NP-hard.
>
> **SayCan Baseline and Affordances (Q9)**
>
> SayCan uses a hand-engineered affordance function to filter infeasible actions. LogicGuard incorporates this naturally: affordance rules are encoded as LTL laws, and the critic adds performance-related constraints based on observed failures. For fairness, SayCan and SayCan+LogicGuard use the same affordance function. InnerMonologue has no affordances, so all feasibility and performance laws are inferred by the critic.

---

> ### Author Response · Authors · 2025-11-18
> **Response to Reviewer 67BB – Summary**
>
> We thank the reviewer for their thoughtful feedback. While we provide detailed responses to each question, we understand that it may be a lot to read. At a high level, we added two ablations/baselines that remove varying levels of LTL and progressively move closer to an LLM-as-a-Judge, which also serves as an online critic mode. This setup performs similarly to LogicGuard on average, but exhibits systematic issues interpreting certain laws and occasionally produces catastrophic failures, reflecting the limitations of blackbox LLMs.
>
> We also thank the reviewer for pointing out the missing standard deviations and confidence intervals; the data was available but was inadvertently omitted before the initial submission. These are now included for all Minecraft experiments.
>
> Finally, we thank the reviewer for their questions regarding design choices and comparisons with LLM-as-a-Judge architectures. Our decisions were guided by standard practices in reinforcement learning and optimization, motivating the actor–critic terminology and the offline critic + symbolic verifier design. Our setup allows LLMs to operate in symbolic, discrete/combinatorial spaces while ensuring consistent, safe, and efficient behavior.

---

### Official Review · Reviewer_jvrz · 2025-10-31

**Soundness:** 3
**Presentation:** 3
**Contribution:** 1
**Rating:** 2
**Confidence:** 4

**Summary:**

The paper proposes LogicGuard, a modular framework designed to augment any embodied LLM agent with a symbolic verification layer. The method consists of an LLM agent which outputs a feasible action and a _critic_ LLM model that analyzes past trajectories to add new laws as Linear Temporal Logic (LTL) from  constraints in natural language, which are then formalized and enforced online by a verifier. They verify empirically their framework on Behavior and Minecraft environments.

**Strengths:**

- The problem addressed by the paper is interesting, they want to automatically add constraints and verify them for embodied LLM agents.
- The integration of LTL-based symbolic constraints with LLM-driven planning is practically interesting.
- The experimental evaluation on two domains (Behavior and Minecraft) demonstrates empirical improvements in completion and efficiency rates.

**Weaknesses:**

- The contribution is primarily empirical rather than methodological. The proposed framework lacks a formal or algorithmic novelty beyond combining existing elements (LLM planners, constraint checking, and symbolic reasoning).
- The entire pipeline is heavily based on LLM, which introduces noise and risk of hallucinating at multiple stages such as: (i) _State grounding_ (ii) _Constraint generation_ which relies on natural language rules induced by an LLM and (iii) _Constraint translation_ where the LLM maps language to atomic propositions, which may not generalize beyond simulator settings.
- While this might work in practice is hard to ensure safety on the constraint generation process. I think there is no automated way to ensure non-redundancy or correctness of generated laws.

**Questions:**

- In a practical scenario how do you enforce safety with respect to generated constraints?
- The approach heavily relies on LLM outputs, how the problem of hallucinations affects this approach?

General comments:
While the idea of symbolic self-supervision for LLM agents is interesting, the proposed method relies to much on LLM accuracy and consequently the contribution feels closer to an empirical demonstration rather than a theoretically or methodologically grounded approach. The experimental results are encouraging but in this current form, the contribution feels limited for a top-tier venue.

---

> ### Author Response · Authors · 2025-11-18
> **Response to Reviewer jvrz**
>
> We would like to thank the reviewer for carefully reading our manuscript and for your feedback. We are glad to hear that the reviewer found our problem setup and our integration of LTL constraints with LLMs interesting. We address the main concerns below.
>
> **W1: Methodological contribution**
>
> We respectfully clarify that LogicGuard is not a static combination of LLM planning and LTL verification. The contribution lies in the algorithmic coupling between the components:
>
> * _Trajectory-conditioned constraint induction._ The critic does not simply output free-form rules; it inspects failed trajectories, identifies recurrent failure modes, and synthesizes reactive LTL constraints anchored to specific transitions. Prior LLM-planning work uses either natural-language heuristics or manually written constraints; LogicGuard provides the first mechanism for data-grounded, formally checkable constraint induction.
>
> * _Closed-loop verifier–critic integration._ Instead of treating symbolic logic as a static safety filter, LogicGuard introduces a feedback loop in which proposed constraints are verified, enforced during replanning, and evaluated for consistency with successful trajectories. This forms an actor–critic update rule over symbolic constraints, without gradient training and without retraining LLMs.
>
> * _Modular, domain-agnostic structure._ The planner, critic, and verifier communicate through a well-defined interface. This structure is not incidental: it allows the same algorithm to transfer from Behavior to Minecraft with no architecture change, and enables independent upgrading or replacement of each module.
>
> Thus, while each building block exists in isolation, the mechanism that connects them is new and leads to capabilities not supported by prior LLM planning frameworks.
>
> **W2/Q2: Hallucinations of LLMs**
>
> We agree that LLM-generated content can be noisy, and LogicGuard explicitly isolates safety from LLM errors.
>
> * Safety-critical rules are hand-specified (or produced by trusted procedures) and enforced exclusively by the symbolic verifier. They are never modified or overridden by the critic.
>
> * The critic can propose only performance-oriented constraints, and these are accepted only if:
>      1) they are grounded in an explicit trajectory fragment,
>      2) they produce well-formed LTL formulas,
>      3) and the verifier can enforce them without violating safe behaviors.
>
> Thus, speculative and syntactically incorrect rules are blocked; while hallucinations may affect performance, they cannot compromise safety, thanks to the strict separation of safety-critical constraints
>
> **W3/Q3: Practical Implementations**
>
> The reviewer raises a valid concern about correctness and redundancy. LogicGuard mitigates these issues through constrained law generation:
>
> * _Generated rules must be reactive LTL formulas._
> This restricts them to local, context-driven adjustments rather than unconstrained global behavior changes.
>
> * _Data grounding prevents unanchored rules._
> Every rule must reference a specific observed transition (e.g., “Trying to make item X, without the right ingredients…”), preventing hallucinated abstractions.
>
> * _Redundancy is benign._
> The verifier is efficient, and redundant constraints do not affect safety or runtime.
>
> * _Incorrect rules are automatically rejected._
> Any proposed constraint that would over-constrain the agent is automatically discarded (and relaxed offline), and rule generation halts when performance degradation is detected.
>
> Overall, these mechanisms establish a structured, auditable framework in which safety-affecting hallucinations are fully suppressed, while any performance-oriented hallucinations remain human-readable, allowing them to be easily identified and corrected without compromising system safety.
>
> **Response to General comments**
>
> We appreciate the reviewer’s perspective and understand the concern regarding empirical vs. methodological contribution. Our work fits into a growing line of research on embodied LLM agents that combine language-driven planning with verifiable control. We respectfully note that the key contribution of LogicGuard is methodological:
>
> * It provides the first symbolic feedback loop for embodied LLM agents, transforming symbolic reasoning from a static filter into an iterative, data-driven supervisory mechanism.
>
> * It establishes a separation between (i) safety, handled entirely by formal verification, and (ii) performance, influenced by automatically generated, yet data-grounded LTL constraints.
>
> * It yields a structured update rule that improves long-horizon behavior without retraining LLMs or requiring differentiable components.
>
> While our empirical results are strong, the framework’s core value lies in its architectural and algorithmic design, which is domain-agnostic, modular, and verifiable.

---

### Author Response · Authors · 2025-12-01
**Summary of Discussion and Changes for AC**

Dear Area Chair,

We sincerely thank the reviewers for their constructive feedback, which has strengthened the paper in terms of novelty, technical rigor, and experimental evaluation. In the following, we summarize the reviewers’ comments, our responses, and the key changes incorporated in the revision.

**Summary of reviews**

Overall, reviewers found the problem timely and interesting and appreciated that LogicGuard combines the generalization capabilities of LLMs with the rigor of symbolic reasoning.

1) Reviewer jvrz: Contribution primarily empirical; concerns about hallucinations
2) Reviewer 67BB: Praised novelty and empirical results; requested more ablations, baselines, and clarification on AP quality, constraint pruning, and comparisons to prior work.
3) Reviewer uyWu: Clarifying questions on architecture and potential rule transfer across environments.
4) Reviewer naM6: Supportive of formal-methods approach and verifier design; raised questions on graph planning and LTL fragment novelty.


**Summary of rebuttal**

1) Reviewer jvrz: We strengthened the contribution and methods sections to highlight our key innovation: using LTL not merely as a safety filter but as a dynamic learning signal that adapts online to the symbolic space. Our self-supervised algorithm mitigates hallucination effects, addressing a major challenge in LLM-based planning.

2) Reviewer 67BB: Added LLM-as-a-judge and LLM-as-a-verifier baselines, with ablations to evaluate trade-offs with standard LLM-judge pipelines. Results show our approach is **more reliable and robust**, particularly in novel environments, we improved theoretical results as suggested and added some related works they mentioned. We also addressed their concerns about sensitivity to AP's in the discussion below.

3) Reviewer uyWu: Clarified offline vs. online critic trade-offs; our experiments show online critics are computationally expensive and more failure-prone. Transfer of rules across tasks remains an exciting future direction.

4) Reviewer naM6: Strengthened discussion of related work and LTL, clearly emphasizing how LogicGuard differs from prior methods both conceptually and technically, highlighting its **novel integration of symbolic verification into LLM planning**.


**Summary of changes**

1) Added LLM-as-a-judge and LLM-as-a-verifier baselines with ablations, showing that LogicGuard improves reliability and robustness over standard LLM-judge approaches. Critically, we note that while LLM-judge architectures misinterpret laws about 11-12% of the time, using our system with formal verification is always correct, and significantly more efficient.
2) Clarified and strengthened the presentation of contributions, highlighting novelty in dynamically using LTL as a learning signal and mitigating hallucinations.
3) Improved positioning and discussion relative to prior work in LTL and formal-methods literature, emphasizing conceptual and methodological novelty.
4) Added a theoretical result bounding the maximum modification an LLM critic can induce on a baseline actor policy, providing formal rigor to complement empirical results.


We thank the reviewers for their thoughtful feedback. We also thank the Area Chair for their time and efforts in reviewing our paper. The revisions clarify and strengthen our contributions, emphasizing the novelty and robustness of LogicGuard. We believe these improvements demonstrate that our work makes a meaningful and impactful contribution to LLM-based planning, overcoming significant drawbacks that such methods have in terms of safety, efficiency, performance and long-term behavior.

---

### Meta-Review · Area_Chair_mepW · 2026-01-02

**Summary:**

I’d like to thank the authors for engaging thoroughly with the reviewer’s detailed feedback and for providing a summary of their rebuttal and changes. At this time, I recommend rejection based on the reviewer’s feedback and my own assessment of the paper. However, I’d like to encourage the authors to continue along this line of work and improve it for a future manuscript; the direction on combining formal methods and LLM planners is a timely one and the initial results are promising.

**jvrz** states that the *strengths* are a good problem setting, practically interesting use of LTL constraints for LLMs, and the empirical results are favorable. The main *concerns* are lack of methodological novelty (the contribution is perceived to be “combining” several existing approaches) and concerns about hallucinations throughout the pipeline.

**67BB** states that the *strengths* are the integration of formal methods into LLMs, strength of empirical results, modular design, and a theorem providing formal guarantees about constraint space complexity. The main *concerns* are the need for additional baselines, evaluation rigor, sensitivity analysis to the atomic propositions (APs), and limited theoretical contributions.

**uyWu** states that the *strengths* are a combination of symbolic reasoning and LLMs, and the interpretability of the generated constraints. The main *concerns* are the expensiveness / instability of the approach and the large amount of hand-designed rules and prompts.

**naM6** states that the *strengths* are the timely research topic, the reasonable proposed pipeline, and the empirical improvements. The main *concerns* are a perceived over-statement of the use of LTL in the work, lack of clarity on the graph planning, lack of novelty on communication via LTL

I’d also like to add recent works in the LLM + robotics + formal methods community, which are relevant for this work and would be helpful to contextualize and potentially use as baselines which directly use formal methods techniques for LLM-base planners:
* [1] ​​Ravichandran, Zachary, et al. "Safety Guardrails for LLM-Enabled Robots." arXiv preprint arXiv:2503.07885 (2025).
* [2] Kapoor, Parv, et al. "Constrained Decoding for Robotics Foundation Models." arXiv preprint arXiv:2509.01728 (2025).
* [3] Wang, Jun, et al. "Conformal temporal logic planning using large language models." ACM Transactions on Cyber-Phys. (2025).

**Reviewer Concerns:**

**jvrz Concerns**
* (1) lack of methodological novelty (the contribution is perceived to be “combining” several existing approaches) $\rightarrow$ **partially addressed (discussion)**. I disagree with the reviewer that doing (rigorous) empirical work to study the combination of existing approaches is not a contribution. That being said, I also believe that the paper could be significantly improved if the problem formulation presented a more rigorous mathematical framework for how LTL verification of LLM planners should modeled. For example, the paper would be strengthened if it framed the problem of LLM planning as a Markov Decision Process (defining each component of what the state is, what actions are, what the transition dynamics are, etc.) and then define how the LTL specification enters into the reward function of this model (or, if it enters as a Q-function for the corresponding MDP). Providing this kind of precise definition of the problem would (a) contribute a formalism for the underlying problem under study, (b) illuminate where approximations / relaxations to the problem are being made, (c) reveal any other necessary baselines or ablations (e.g., increasing the partial observability of the states required for verification). Note: in the rebuttal the authors state “Extending to full LTL would require predicting how actions affect future observations. This would necessitate either a learned world model (introducing model error and hallucination risks) or explicit environment simulators (limiting generality).” I think this is a great aspect to introspect on, as it could make the proposed work stronger. It also has a tight connection to the MDP formulation I proposed above, since it makes it clear how the planning system needs the ability to forward simulate how different actions can cause different safe/unsafe outcomes.
* (2) concerns about hallucinations throughout the pipeline $\rightarrow$ **addressed**.

**67BB Concerns**
* (1) additional baselines $\rightarrow$ **addressed (empirically)**. Added LLM-as-a-judge and LLM-as-a-verifier baselines showing similar mean results to the LTL approach, but with larger computational overhead and higher variance.
* (2) evaluation rigor $\rightarrow$ **addressed (empirically)**. Added more comprehensive statistical reporting (mean, std, and 95% confidence intervals).
* (3) sensitivity analysis to the atomic propositions (APs) $\rightarrow$ **partially addressed (discussion)**.
* (4) limited theoretical contributions $\rightarrow$ **addressed**. Renamed Theorem 1 to Remark 1.

**uyWu Concerns**
* (1) expensiveness / instability of the approach $\rightarrow$ **addressed**. It would have been even more helpful to provide the concrete numbers on the latency and the performance in the rebuttal.
* (2) large amount of hand-designed rules and prompts $\rightarrow$ **partially addressed (discussion)**.

**naM6 Concerns**
* (1) over-statement of the use of LTL in the work $\rightarrow$ **addressed (updated manuscript to soften claims)**.
* (2) lack of clarity on the graph planning $\rightarrow$ **addressed (discussion)**.
* (3) lack of novelty on communication via LTL $\rightarrow$ **not addressed**.

**Reviewer Scores:**

* **jvrz** would have *maintained* a score of 2: reject, not good enough
* **67BB** would have *maintained* a 6: marginally above the acceptance threshold.
* **uyWu** would have *maintained* a score of 4: marginally below the acceptance threshold.
* **naM6** would have *maintained* a score of 6: marginally above the acceptance threshold.

---

### Decision · Program_Chairs · 2026-01-26

Reject